# Non-Gaussian Gaussian Processes for Few-Shot Regression

**Marcin Sendera** *
Jagiellonian University

**Jacek Tabor**
Jagiellonian University

**Aleksandra Nowak**
Jagiellonian University

**Andrzej Bedychaj**
Jagiellonian University

**Massimiliano Patacchiola**
University of Cambridge

**Tomasz Trzcinski**
Jagiellonian University,
Warsaw University of Technology,
Tooploox

**Przemysław Spurek**
Jagiellonian University

**Maciej Zieba**
Wrocław University of
Science and Technology,
Tooploox

## Abstract

Gaussian Processes (GPs) have been widely used in machine learning to model distributions over functions, with applications including multi-modal regression, time-series prediction, and few-shot learning. GPs are particularly useful in the last application since they rely on Normal distributions and enable closed-form computation of the posterior probability function. Unfortunately, because the resulting posterior is not flexible enough to capture complex distributions, GPs assume high similarity between subsequent tasks – a requirement rarely met in real-world conditions. In this work, we address this limitation by leveraging the flexibility of Normalizing Flows to modulate the posterior predictive distribution of the GP. This makes the GP posterior locally non-Gaussian, therefore we name our method Non-Gaussian Gaussian Processes (NGGPs). We propose an invertible ODE-based mapping that operates on each component of the random variable vectors and shares the parameters across all of them. We empirically tested the flexibility of NGGPs on various few-shot learning regression datasets, showing that the mapping can incorporate context embedding information to model different noise levels for periodic functions. As a result, our method shares the structure of the problem between subsequent tasks, but the contextualization allows for adaptation to dissimilarities. NGGPs outperform the competing state-of-the-art approaches on a diversified set of benchmarks and applications.

## 1 Introduction

Gaussian Processes (GPs) [33, 46] are one of the most important probabilistic methods, and they have been widely used to model distributions over functions in a variety of applications such as multi-modal regression [56], time-series prediction [3, 27] and meta-learning [29, 45]. Recent works propose to use GPs in the *few-shot learning* scenario [4, 29, 39, 49], where the model is trained to solve a supervised task with only a few labeled samples available. This particular application is well-fitted to GPs since they can determine the posterior distribution in closed-form from a small set of data samples [29].

---

*Corresponding author: `marcin.sendera@doctoral.uj.edu.pl`

35th Conference on Neural Information Processing Systems (NeurIPS 2021).

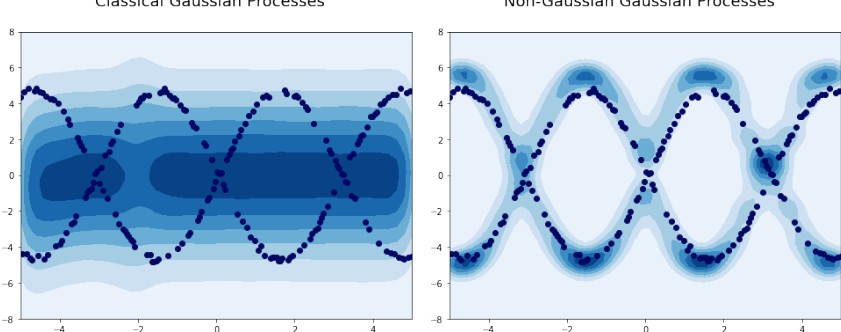

Figure 1: Results of Deep Kernels with classical GP (left) and NGGP (right). The one-dimensional samples were generated randomly from $\sin(x)$ and $-\sin(x)$ functions with additional noise. NGGP, compared to GP, does not have an assumption of Gaussian prior, which allows for modeling a multi-modal distribution.

However, the generalization capabilities of GPs come at the price of reduced flexibility when the modeled distributions are complex, *e.g.*, they have high skewness or heavy tails. Furthermore, GPs assume a high similarity between subsequent tasks. This condition is rarely met in real-world applications where tasks can vary during time, as is the case in heteroscedastic regression. These limitations of GPs also extend to multi-modal learning or, more generally, to multi-label regression [56].

In this work, we address those drawbacks by modeling the GPs posterior predictive distributions with a local non-Gaussian approximation. We do so by introducing a new method that we have named *Non-Gaussian Gaussian Processes (NGGPs)*. In NGGPs, we leverage the flexibility of Continuous Normalizing Flows (CNF) [16] to model arbitrary probability distributions. In particular, we propose an invertible ODE-based mapping that operates on each component of the random variable vectors. This way, we can compute a set of CNFs parameters shared across all vectors, with the resulting mapping incorporating the information of the context to model different noise for periodic functions. Figure 1 shows how NGGPs are able to capture the overall structure of a problem, whereas standard GPs fail. NGGPs are able to reconstruct a multi-modal sine function while adapting to local dissimilarities thanks to the contextualization provided by the ODE-based mapping. We provide empirical evidence that NGGPs outperform competitive state-of-the-art approaches on a diversified set of benchmarks and applications in a few-shot learning scenario; the code is released with an open-source license[2].

The contributions of our work can be summarized as follows:

- We introduce Non-Gaussian Gaussian Processes (NGGPs), a new probabilistic method for modeling complex distributions through locally non-Gaussian posteriors.
- We show how invertible ODE-based mappings can be coupled with GPs to process the marginals of multivariate random variables resulting in more flexible models.
- We extensively test NGGPs on a variety of few-shot learning benchmarks, achieving state-of-the-art performances in most conditions.

## 2  Related Work

The related work section is divided into three parts. First, we present a general Few-Shot Learning problem. Then, we discuss GPs, focusing on models, which use flow architectures. Finally, in the third paragraph, we describe existing approaches to Few-Shot Learning, which use Gaussian Processes.

**Few-Shot Learning**  Few-Shot Learning aims at solving problems in which the number of observations is limited. Some of the early methods in this domain have applied a two-phase approach by pre-training on the base set of training tasks and then fine-tuning the parameters to the test tasks [4, 28]. An alternative approach is given by non-parametric metric-learning algorithms, which aim at optimizing a metric, that is then used to calculate the distance between the target observations

---

[2]https://github.com/gmum/non-gaussian-gaussian-processes

and the support set items [48, 38, 42]. Another popular approach to few-shot learning is Model Agnostic Meta-Learning (MAML) [9] and its variants [12, 24, 32, 54, 14, 52, 6]. MAML aims at finding a set of joined task parameters that can be easily fine-tuned to new test tasks via few gradient descent updates. MAML can also be treated as a Bayesian hierarchical model [10, 15, 18]. Bayesian MAML [55] combines efficient gradient-based meta-learning with non-parametric variational inference in a principled probabilistic framework. A few algorithms have been focusing exclusively on regression tasks. An example is given by ALPaCA [17], which uses a dataset of sample functions to learn a domain-specific encoding and prior over weights.

**Gaussian Processes** GPs have been applied to numerous machine learning problems, such as spatio-temporal density estimation [7], robotic control [53], or dynamics modeling in transcriptional processes in the human cell [21]. The drawback of GP lies in the computational cost of the training step, which is $O(n^3)$ (where $n$ denotes the number of observations in the training sample).

In [41], the authors extend the flexibility of GPs by processing the targets with a learnable monotonic mapping (the warping function). This idea is further extended in [22], which shows that it is possible to place the prior of another GP on the warping function itself. Our method is different from these approaches, since the likelihood transformation is obtained by the use of a learnable CNF mapping.

In [26], the authors present the Transformed Gaussian Processes (TGP), a new flexible family of function priors that use GPs and flow models. TGPs exploit Bayesian Neural Networks (BNNs) as input-dependent parametric transformations. The method can match the performance of Deep GPs at a fraction of the computational cost.

The methods discussed above are trained on a single dataset, that is kept unchanged. Therefore, it is not trivial to adapt such methods to the the few-shot setting.

**Few-Shot Learning with Gaussian Processes** When the number of observations is relatively small, GPs represent an interesting alternative to other regression approaches. This makes GPs a good candidate for meta-learning and few-shot learning, as shown by recent publications that have explored this research direction. For instance, Adaptive Deep Kernel Learning (ADKL) [45] proposes a variant of kernel learning for GPs, which aims at finding appropriate kernels for each task during inference by using a meta-learning approach. A similar approach can be used to learn the mean function [11]. In [37], the authors presented a theoretically principled PAC-Bayesian framework for meta-learning. It can be used with different base learners (e.g., GPs or BNNs). Topics related to kernel tricks and meta-learning have been explored in [47]. The authors propose to use nonparametric kernel regression for the inner loop update. In [43], the authors introduce an information-theoretic framework for meta-learning by using a variational approximation to the information bottleneck. In their GP-based approach, to account for likelihoods other than Gaussians, they propose approximating the non-Gaussian terms in the posterior with Gaussian distributions (by using amortized functions), while we use CNFs to increase the flexibility of the GPs.

In [29], the authors present Deep Kernel Transfer (DKT): a Bayesian treatment for the meta-learning inner loop through the use of deep kernels, which has achieved state-of-the-art results. In DKT, the deep kernel and the parameters of the GP are shared across all tasks and adjusted to maximize the marginal log-likelihood, which is equivalent to Maximum-Likelihood type II (ML-II) learning. DKT is particularly effective in the regression case since it is able to capture prior knowledge about the data through the GP kernel. However, in many settings, prior assumptions could be detrimental if they are not met during the evaluation phase. This is the case in few-shot regression, where there can be a significant difference between the tasks seen at training time and the tasks seen at evaluation time. For instance, if we are given few-shot tasks consisting of samples from periodic functions but periodicity is violated at evaluation time, then methods like DKT may suffer in terms of predictive accuracy under this domain shift. In this work, we tackle this problem by exploiting the flexibility of CNFs.

## 3 Background

**Gaussian Processes.** The method proposed in this paper strongly relies on Gaussian Processes (GPs) and their applications in regression problems. GPs are a well-established framework for principled uncertainty quantification and automatic selection of hyperparameters through a marginal likelihood objective [35]. More formally, a GP is a collection of random variables such that the joint distribution of every finite subset of random variables from this collection is a multivariate Gaussian [31]. We

denote Gaussian Process as $f(\cdot) \sim \mathcal{GP}(\mu(\cdot), k(\cdot, \cdot))$, where $\mu(\mathbf{x})$ and $k(\mathbf{x}, \mathbf{x}')$ are the mean and covariance functions. When prior information is not available, a common choice for $\mu$ is the zero constant function. The covariance function must impose a valid covariance matrix. This is achieved by restricting $k$ to be a kernel function. Examples of such kernels include the Linear kernel, Radial Basis Function (RBF) kernel, Spectral Mixture (Spectral) kernel [50], or Cosine-Similarity kernel [33]. Kernel functions can also be directly modeled as inner products defined in the feature space imposed by a feature mapping $\psi : X \to V$:

$$k(x, x') = \langle \psi(x), \psi(x') \rangle_V \tag{1}$$

An advantage of the formulation above is that it can be easily implemented by modeling $\psi$ through a neural network. Throughout this work, we call this technique the NN Linear kernel (sometimes called Deep Kernel [29]). Since every kernel can be described in terms of Equation (1), such an approach may be desired if no prior information about the structure of the kernel function is available.

Gaussian Processes provide a method for modeling probability distributions over functions. Consider a regression problem:

$$\mathbf{y}_i = f(\mathbf{x}_i) + \epsilon_i, \text{ for } i = 1, \ldots, m, \tag{2}$$

where $\epsilon_i$ are i.i.d. noise variables with independent $\mathcal{N}(0, \sigma^2)$ distributions. Let $\mathbf{X}$ be the matrix composed of all samples $\mathbf{x}_i$ and let $\mathbf{y}$ be the vector composed of all target values $\mathbf{y}_i$. Assuming that $f(\cdot) \sim \mathcal{GP}(0, k(\cdot, \cdot))$, we obtain:

$$\mathbf{y}|\mathbf{X} \sim \mathcal{N}(0, \mathbf{K} + \sigma \mathbb{I}), \tag{3}$$

where $k_{i,j} = k(\mathbf{x}_i, \mathbf{x}_j)$. Analogously, inference over the unknown during the training samples is obtained by conditioning over the normal distribution. Let $(\mathbf{y}, \mathbf{X})$ be the train data and let $(\mathbf{y}_*, \mathbf{X}_*)$ be the test data. Then the distribution of $\mathbf{y}_*$ given $\mathbf{y}, \mathbf{X}, \mathbf{X}_*$ is also a Gaussian distribution [34]:

$$\mathbf{y}_*|\mathbf{y}, \mathbf{X}, \mathbf{X}_* \sim \mathcal{N}(\mu_*, \mathbf{K}_*), \tag{4}$$

where:

$$\boldsymbol{\mu}_* = \mathbf{K}(\mathbf{X}_*, \mathbf{X})\left(\mathbf{K}(\mathbf{X}, \mathbf{X}) + \sigma^2\mathbb{I}\right)^{-1}\mathbf{y}$$

$$\mathbf{K}_* = \mathbf{K}(\mathbf{X}_*, \mathbf{X}_*) + \sigma^2\mathbb{I} - \mathbf{K}(\mathbf{X}_*, \mathbf{X})\left(\mathbf{K}(\mathbf{X}, \mathbf{X}) + \sigma^2\mathbb{I}\right)^{-1}\mathbf{K}(\mathbf{X}, \mathbf{X}_*)$$

**Continuous Normalizing Flows.** Normalizing Flows (NF) [36] are gaining popularity among generative models thanks to their flexibility and the ease of training via direct negative log-likelihood (NLL) optimization. Flexibility is given by the change-of-variable technique that maps a latent variable $\mathbf{z}$ with know prior $p(\mathbf{z})$ to $\mathbf{y}$ from some observed space with unknown distribution. This mapping is performed through a series of (parametric) invertible functions: $\mathbf{y} = \mathbf{f}_n \circ \cdots \circ \mathbf{f}_1(\mathbf{z})$. Assuming known prior $p(\mathbf{z})$ for $\mathbf{z}$, the log-likelihood for $\mathbf{y}$ is given by:

$$\log p(\mathbf{y}) = \log p(\mathbf{z}) - \sum_{n=1}^{N} \log \left| \det \frac{\partial \mathbf{f}_n}{\partial \mathbf{z}_{n-1}} \right|, \tag{5}$$

where $\mathbf{z} = \mathbf{f}_1^{-1} \circ \cdots \circ \mathbf{f}_n^{-1}(\mathbf{y})$ is a result of the invertible mapping. The biggest challenge in normalizing flows is the choice of the invertible functions $\mathbf{f}_n, \ldots, \mathbf{f}_1$. This is due to the fact that they need to be expressive while guaranteeing an efficient calculation of the Jacobian determinant, which usually has a cubic cost. An alternative approach is given by CNF models [16]. CNFs use continuous, time-dependent transformations instead of sequence of discrete functions $\mathbf{f}_n, \ldots, \mathbf{f}_1$. Formally, we introduce a function $\mathbf{g}_{\boldsymbol{\beta}}(\mathbf{z}(t), t)$ that models the dynamics of $\mathbf{z}(t)$, $\frac{\partial \mathbf{z}(t)}{\partial t} = \mathbf{g}_{\boldsymbol{\beta}}(\mathbf{z}(t), t)$, parametrized by $\boldsymbol{\beta}$. In the CNF setting, we aim at finding a solution $\mathbf{y} := \mathbf{z}(t_1)$ for the differential equation, assuming the given initial state $\mathbf{z} := \mathbf{z}(t_0)$ with a known prior. As a consequence, the transformation function $\mathbf{f}_{\boldsymbol{\beta}}$ is defined as:

$$\mathbf{y} = \mathbf{f}_{\boldsymbol{\beta}}(\mathbf{z}) = \mathbf{z} + \int_{t_0}^{t_1} \mathbf{g}_{\boldsymbol{\beta}}(\mathbf{z}(t), t)dt. \tag{6}$$

The inverted form of the transformation can be easily computed using the formula:

$$\mathbf{f}_{\boldsymbol{\beta}}^{-1}(\mathbf{y}) = \mathbf{y} - \int_{t_0}^{t_1} \mathbf{g}_{\boldsymbol{\beta}}(\mathbf{z}(t), t)dt. \tag{7}$$

The log-probability of $\mathbf{y}$ can be computed by:

$$\log p(\mathbf{y}) = \log p(\mathbf{f}_{\boldsymbol{\beta}}^{-1}(\mathbf{y})) - \int_{t_0}^{t_1} \text{Tr}\left(\frac{\partial \mathbf{g}_{\boldsymbol{\beta}}}{\partial \mathbf{z}(t)}\right) dt \quad \text{where} \quad \mathbf{f}_{\boldsymbol{\beta}}^{-1}(\mathbf{y}) = \mathbf{z}. \tag{8}$$

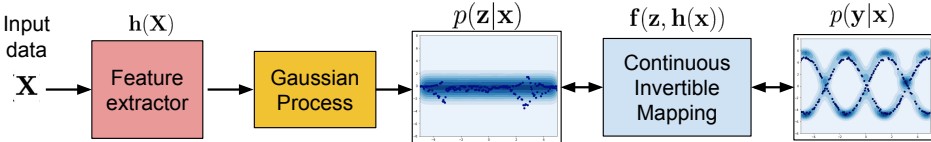

Figure 3: The general architecture of our approach. The input data are embedded by the feature extractor $\mathbf{h}(\cdot)$ and then used to create a kernel for the GP. Next, the output $\mathbf{z}$ of the GP is adjusted using an invertible mapping $\mathbf{f}(\cdot)$ which is conditioned on the output of the feature extractor. This allows us to model complex distributions of the target values $\mathbf{y}$.

## 4 Non-Gaussian Gaussian Processes

In this work, we introduce Non-Gaussian Gaussian Processes (NGGPs) to cope with the significant bottlenecks of Gaussian Processes for Few-Shot regression tasks: reduced flexibility and assumption about the high similarity between the structure of subsequent tasks. We propose to model the posterior predictive distribution as non-Gaussian on each datapoint. We are doing so by incorporating the flexibility of CNFs. However, we do not stack the CNF on GP to model the multidimensional distribution over $\mathbf{y}$. Instead, we attack the problem with an invertible ODE-based mapping that can utilize each component of the random variable vector and create the specific mapping for each datapoint (see Figure 2).

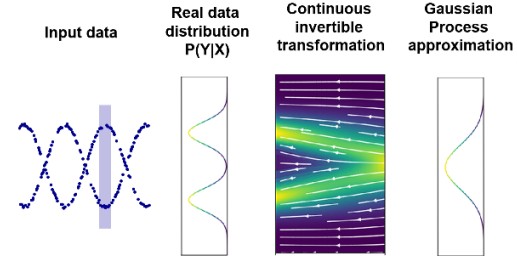

Figure 2: General idea of NGGP. A complex multi-modal distribution can be modelled by exploiting a continuous invertible transformation to fit the Normal distribution used by the GP. Image inspired by Figure 1 in [16].

The general overview of our method is presented in Figure 3. Consider the data matrix $\mathbf{X}$, which stores the observations $\mathbf{x}_i$ for a given task. Each element is processed by a feature extractor $\mathbf{h}(\cdot)$ to create the latent embeddings. Next, we model the distribution of the latent variable $\mathbf{z}$ with a GP. Further, we use an invertible mapping $\mathbf{f}(\cdot)$ in order to model more complex data distributions. Note that the transformation is also conditioned on the output of the feature extractor $\mathbf{h}(\cdot)$ to include additional information about the input.

The rest of this section is organized as follows. In Section 4.1, we demonstrate how the marginal can be calculated during training. In Section 4.2, we demonstrate how to perform an inference stage with the model. Finally, in Section 4.3, we show how the model is applied to the few-shot setting.

### 4.1 Training objective

Consider the GP with feature extractor $\mathbf{h}_{\boldsymbol{\phi}}(\cdot)$ parametrized by $\boldsymbol{\phi}$ and any kernel function $k_{\boldsymbol{\theta}}(\cdot, \cdot)$ parametrized by $\boldsymbol{\theta}$. Assuming the given input data $\mathbf{X}$ and corresponding output values $\mathbf{z}$, we can define the marginal log-probability for the GP:

$$\log p(\mathbf{z}|\mathbf{X}, \boldsymbol{\phi}, \boldsymbol{\theta}) = -\frac{1}{2}\mathbf{z}^{\mathrm{T}}(\mathbf{K} + \sigma^2 \mathbb{I})^{-1}\mathbf{z} - \frac{1}{2}\log|\mathbf{K} + \sigma^2 \mathbb{I}| - \frac{D}{2}\log(2\pi), \qquad (9)$$

where $D$ is the dimension of $\mathbf{y}$, $\mathbf{K}$ is the kernel matrix, and $k_{i,j} = k_{\boldsymbol{\theta}}(\mathbf{h}_{\boldsymbol{\phi}}(\mathbf{x}_i), \mathbf{h}_{\boldsymbol{\phi}}(\mathbf{x}_j))$.

Taking into account Equation (8) we can express the log marginal likelihood as follows:

$$\log p(\mathbf{y}|\mathbf{X}, \boldsymbol{\phi}, \boldsymbol{\theta}, \boldsymbol{\beta}) = \log p(\mathbf{z}|\mathbf{X}, \boldsymbol{\phi}, \boldsymbol{\theta}) - \int_{t_0}^{t_1} \mathrm{Tr}\left(\frac{\partial \mathbf{g}_{\boldsymbol{\beta}}}{\partial \mathbf{z}(t)}\right) dt, \qquad (10)$$

where $\mathbf{f}_{\boldsymbol{\beta}}^{-1}(\mathbf{y}) = \mathbf{z}$, $p(\mathbf{z}|\mathbf{X}, \boldsymbol{\phi}, \boldsymbol{\theta})$ is the marginal defined by Equation (9) and $\mathbf{f}_{\boldsymbol{\beta}}^{-1}(\cdot)$ is the transformation given by Equation (6). In the next stage of the pipeline, we propose to apply the flow transformation $\mathbf{f}_{\boldsymbol{\beta}}^{-1}(\cdot)$ independently to each one of the marginal elements in $\mathbf{y}$, that is $\mathbf{f}_{\boldsymbol{\beta}}^{-1}(\mathbf{y}) = [f_{\boldsymbol{\beta}}^{-1}(y_1), \ldots, f_{\boldsymbol{\beta}}^{-1}(y_D)]^{\mathrm{T}}$, with $f_{\boldsymbol{\beta}}^{-1}(\cdot)$ sharing its parameters across all components. In other words, while the GP captures the dependency across the variables, the flow operates independently on the marginal components of $\mathbf{y}$. Additionally, the flow is conditioned on the information

**Algorithm 1** NGGP in the few-shot setting, train and test functions.

---

**Require:** $\mathcal{D} = \{\mathcal{T}_n\}_{n=1}^N$ train dataset and $\mathcal{T}_* = \{\mathcal{S}_*, \mathcal{Q}_*\}$ test task.
**Parameters:** $\boldsymbol{\theta}$ kernel hyperparameters, $\boldsymbol{\phi}$ feature extractor parameters, $\boldsymbol{\beta}$ flow transformation parameters.
**Hyperparameters:** $\alpha, \eta, \gamma$: step size hyperparameters for the optimizers.

 1: **function** TRAIN($\mathcal{D}, \alpha, \eta, \gamma, \boldsymbol{\theta}, \boldsymbol{\phi}, \boldsymbol{\beta}$)
 2:     **while** not done **do**
 3:         Sample task $\mathcal{T} = (\mathbf{X}, \mathbf{y}) \sim \mathcal{D}$
 4:         $\mathcal{L} = -\log p(\mathbf{y}|\mathbf{X}, \boldsymbol{\theta}, \boldsymbol{\phi}, \boldsymbol{\beta})$                             ▷ See Equation (13)
 5:         Update $\boldsymbol{\theta} \leftarrow \boldsymbol{\theta} - \alpha\nabla_{\theta}\mathcal{L}$,             ▷ Updating kernel hyperparameters
 6:                $\boldsymbol{\phi} \leftarrow \boldsymbol{\phi} - \eta\nabla_{\phi}\mathcal{L}$,          ▷ Updating feature extractor parameters
 7:                $\boldsymbol{\beta} \leftarrow \boldsymbol{\beta} - \gamma\nabla_{\beta}\mathcal{L}$        ▷ Updating flow transformation parameters
 8:     **end while**
 9:     **return** $\boldsymbol{\theta}, \boldsymbol{\phi}, \boldsymbol{\beta}$
10: **end function**

11: **function** TEST($\mathcal{T}_*, \boldsymbol{\theta}, \boldsymbol{\phi}, \boldsymbol{\beta}$)
12:     Assign support $\mathcal{S}_* = (\mathbf{X}_{*,s}, \mathbf{y}_{*,s})$ and query $\mathcal{Q}_* = (\mathbf{X}_{*,q}, \mathbf{y}_{*,q})$
13:     **return** $p(\mathbf{y}_{*,q}|\mathbf{X}_{*,q}, \mathbf{y}_{*,s}, \mathbf{X}_{*,s},, \boldsymbol{\theta}, \boldsymbol{\phi}, \boldsymbol{\beta})$          ▷ See Equation (14)
14: **end function**

---

encoded by the feature extractor, such that it can account for the context information $\mathbf{h}_{\phi}(\mathbf{x}_d)$ from the corresponding input value $\mathbf{x}_d$:

$$y_d = f_{\boldsymbol{\beta}}(z_d, \mathbf{h}_{\phi}(\mathbf{x}_d)) = z_d + \int_{t_0}^{t_1} g_{\boldsymbol{\beta}}(z_d(t), t, \mathbf{h}_{\phi}(\mathbf{x}_d))dt. \tag{11}$$

The inverse transformation can be easily calculated with the following formula:

$$f_{\boldsymbol{\beta}}^{-1}(y_d) = y_d - \int_{t_0}^{t_1} g_{\boldsymbol{\beta}}(z_d(t), t, \mathbf{h}_{\phi}(\mathbf{x}_d))dt \tag{12}$$

The final marginal log-likelihood can be expressed as:

$$\log p(\mathbf{y}|\mathbf{X}, \boldsymbol{\phi}, \boldsymbol{\theta}, \boldsymbol{\beta}) = \log p(\mathbf{z}^{\mathbf{h}}|\mathbf{X}, \boldsymbol{\phi}, \boldsymbol{\theta}) - \sum_{d=1}^{D} \int_{t_0}^{t_1} \frac{\partial g_{\boldsymbol{\beta}}}{\partial z_d(t)}dt, \tag{13}$$

where $\mathbf{z}^{\mathbf{h}} = \mathbf{f}_{\boldsymbol{\beta}}^{-1}(\mathbf{y}, \mathbf{h}_{\phi}(\mathbf{X}))$ is the vector of inverse functions $f_{\boldsymbol{\beta}}(z_d, \mathbf{h}_{\phi}(\mathbf{x}_d))$ given by Equation (12).

The transformation described above can be paired with popular CNF models. Here we choose Ffjord [16], which has showed to perform better on low-dimensional data when compared against discrete flows like RealNVP [5] or Glow [19]. Note that, the CNF is applied independently on the components of the GP outputs and shared across them. Therefore, we do not have any issue with the estimation of the Jacobian, since this corresponds to the first-order derivative of the output w.r.t. the scalar input.

### 4.2 Inference with the model

At inference time, we estimate the posterior predictive distribution $p(\mathbf{y}_*|\mathbf{X}_*, \mathbf{y}, \mathbf{X}, \boldsymbol{\phi}, \boldsymbol{\theta}, \boldsymbol{\beta})$, where we have access to training data $(\mathbf{y}, \mathbf{X})$ and model the probability of $D_*$ test outputs $\mathbf{y}_*$ given the inputs $\mathbf{X}_*$. The posterior has a closed expression (see Section 3). Since the transformation given by Equation (11) operates independently on the outputs, we are still able to model the posterior in closed form:

$$\log p(\mathbf{y}_*|\mathbf{X}_*, \mathbf{y}, \mathbf{X}, \boldsymbol{\phi}, \boldsymbol{\theta}, \boldsymbol{\beta}) = \log p(\mathbf{z}_*^{\mathbf{h}}|\mathbf{X}, \mathbf{z}^{\mathbf{h}}, \mathbf{X}, \boldsymbol{\phi}, \boldsymbol{\theta}) - \sum_{d=1}^{D_*} \int_{t_0}^{t_1} \frac{\partial g_{\boldsymbol{\beta}}}{\partial z_d(t)}dt, \tag{14}$$

where $\mathbf{z}_*^{\mathbf{h}} = f_{\boldsymbol{\beta}}^{-1}(\mathbf{y}_*, \mathbf{h}_{\phi}(\mathbf{X}_*))$, $\mathbf{z}^{\mathbf{h}} = f_{\boldsymbol{\beta}}^{-1}(\mathbf{y}, \mathbf{h}_{\phi}(\mathbf{X}))$ are the inverted transformations for test and train data, and $p(\mathbf{z}_*^{\mathbf{h}}|\mathbf{X}_*, \mathbf{z}^{\mathbf{h}}, \mathbf{X}, \boldsymbol{\phi}, \boldsymbol{\theta})$ is the GP posterior described in Equation (4).

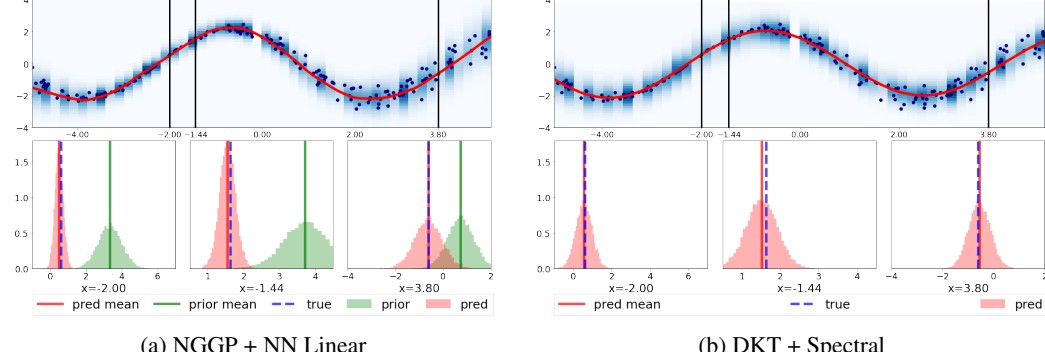

|  |  |
|---|---|
| (a) NGGP + NN Linear | (b) DKT + Spectral |

Figure 4: The results for the sines dataset with mixed-noise for the best performing kernels for NGGP (NN Linear) and DKT (Spectral). The top plot in each figure represents the estimated density (blue hue) and predicted curve (red line), as well as the true test samples (navy blue dots). For three selected input points (denoted by black vertical lines), we plot the obtained marginal densities in the bottom images (red color). In addition, for the NGGP method, we also plot the marginal priors (in green) for each of these three points. It may be observed that NGGP is more successful in modeling the marginal for varying noise levels.

## 4.3 Adaptation for few-shot regression

In few-shot learning, we are given a meta-dataset of tasks $\mathcal{D} = \{\mathcal{T}_n\}_{n=1}^N$ where each task $\mathcal{T}_n$ contains a support set $\mathcal{S}_n$, and a query set $\mathcal{Q}_n$. At training time, both support and query contain input-output pairs $(\mathbf{X}, \mathbf{y})$, and the model is trained to predict the target in the query set given the support. At evaluation time, we are given a previously unseen task $\mathcal{T}_* = (\mathcal{S}_*, \mathcal{Q}_*)$, and the model is used to predict the target values of the unlabeled query points. We are interested in few-shot regression, where inputs are vectors and outputs are scalars.

We follow the paradigm of Deep Kernel Transfer (DKT) introduced in [29] and propose the following training and testing procedures (see Algorithm 1). During the training stage, we randomly sample the task, calculate the loss defined by Equation (13) and update all the parameters using gradient-based optimization. During testing, we simply identify the query and support sets and calculate the posterior given by Equation (14).

## 5 Experiments

In this section, we provide an extensive evaluation of our approach (NGGP) on a set of challenging few-shot regression tasks. We compare the results with other baseline methods used in this domain. As quantitative measures, we use the standard mean squared error (*MSE*) and, when applicable, the negative log-likelihood (*NLL*).

**Sines dataset** We start by comparing NGGP to other few-shot learning algorithms in a simple regression task defined on sines functions. To this end, we adapt the dataset from [9] in which every task is composed of points sampled from a sine wave with amplitude in the range $[0.1, 5.0]$, phase in the range $[0, \pi]$, and Gaussian noise $\mathcal{N}(0, 0.1)$. The input points are drawn uniformly at random from the range $[-5, 5]$. We consider 5 support and 5 query points during the training and 5 support and 200 query points during inference. In addition, following [29], we also consider an *out-of-range* scenario, in which the range during the inference is extended to $[-5, 10]$. We also perform a variation of sines experiment in which we inject input-dependent noise. The target values in this setting are modeled by $A \sin(x + \varphi) + |x + \varphi|\epsilon$, where the amplitude, phase, input, and noise points are drawn from the same distributions as in the standard setup described before. We refer to this dataset ablation as *mixed-noise sines*. For more information about the training regime and architecture, refer to Supplementary Materials A. Table 1 presents the results of the experiments. We use the DKT method as a reference since it provides state-of-the-art results for the few-shot sines dataset [29]. For a report with more baseline methods, please refer to Supplementary Materials B.

Both DKT and our NGGP perform very well when paired with the Spectral Mixture Kernel, achieving the same performance on *in-range* data. However, our approach gives superior results in the *out-of-*

Table 1: The *MSE* and *NLL* results for the inference tasks on sines datasets in the *in-range* and *out-range* settings. Lowest results in bold (the lower the better).

| Method | sines | | | | mixed-noise sines | | | |
| --- | --- | --- | --- | --- | --- | --- | --- | --- |
| | in-range | | out-of-range | | in-range | | out-of-range | |
| | *MSE* | *NLL* | *MSE* | *NLL* | *MSE* | *NLL* | *MSE* | *NLL* |
| DKT + RBF | 1.36±1.64 | -0.76±0.06 | 2.94±2.70 | -0.69±0.06 | 1.60±1.63 | 0.48 ± 0.22 | 2.99± 2.37 | 2.01 ± 0.59 |
| DKT + Spectral | **0.02±0.01** | **-0.83±0.03** | 0.04±0.03 | -0.70±0.14 | **0.18 ± 0.12** | 0.37±0.16 | 1.33 ± 1.10 | 1.58 ± 0.40 |
| DKT + NN Linear | **0.02±0.02** | -0.73±0.11 | 6.61±31.63 | 38.38±40.16 | **0.18±0.11** | 0.45 ± 0.23 | 5.85 ± 12.10 | 8.64 ± 6.55 |
| NGGP + RBF | 1.02±1.40 | -0.74±0.07 | 3.02±2.53 | -0.65±0.08 | 1.30±1.36 | 0.33 ± 0.16 | 3.90 ± 2.60 | 1.83 ± 0.53 |
| NGGP + Spectral | **0.02±0.01** | **-0.83±0.05** | **0.03±0.02** | **-0.80±0.07** | 0.22 ± 0.14 | 0.44 ± 0.19 | **1.14 ± 0.90** | **1.35 ± 0.38** |
| NGGP + NN Linear | 0.04±0.03 | -0.73±0.10 | 7.34±12.85 | 29.86±27.97 | 0.20 ± 0.12 | **0.17 ± 0.15** | 4.74 ± 6.29 | 2.92 ± 1.93 |

*range* scenario, confirming that NGGP is able to provide a better estimate of the predictive posterior for the unseen portions of the task. It is also worth noting that in all settings, NGGP consistently achieves the best *NLL* results. This is particularly evident for the *in-range* mixed-noise sines dataset. We analyze this result in Figure 4, where NGGP successfully models the distribution of the targets, predicting narrow marginals for the more centralized points and using wider distributions for the points with larger noise magnitude. This is in contrast with DKT, which fails to capture different noise levels within the data. These observations confirm our claim that the NGGP is able to provide a good estimate in the case of heteroscedastic data.

**Head-pose trajectory** In this experiment, we use the Queen Mary University of London multiview face dataset [13]. This dataset is composed of grayscale face images of 37 people (32 train, 5 test). There are 133 facial images per person, covering a viewsphere of $\pm 90°$ in yaw and $\pm 30°$ in tilt at $10°$ increment. We follow the evaluation procedure provided in [29]. Each task consists of randomly sampled trajectories taken from this discrete manifold. The *in-range* scenario includes the full manifold, while the *out-of-range* scenario includes only the leftmost 10 angles. At evaluation time, the inference is performed over the full manifold with the goal of predicting the tilt. The results are provided in Table 2. In terms of *MSE*, our NGGP method is competitive with other approaches, but it achieves significantly better *NLL* results, especially in the *out-of-range* setting. This suggests that NGGPs are indeed able to adapt to the differences between the tasks seen at training time and tasks seen at evaluation time by providing a probability distribution that accurately captures the true underlying data.

Table 2: Quantitative results for Queen Mary University of London for *in-range* and *out-of-range* settings, taking into account *NLL* and *MSE* measures.

| Method | in-range | | out-of-range | |
| --- | --- | --- | --- | --- |
| | *MSE* | *NLL* | *MSE* | *NLL* |
| Feature Transfer/1 | 0.25±0.04 | - | 0.20±0.01 | - |
| Feature Transfer/100 | 0.22±0.03 | - | 0.18±0.01 | - |
| MAML (1 step) | 0.21±0.01 | - | 0.18±0.02 | - |
| DKT + RBF | 0.12±0.04 | 0.13±0.14 | 0.14±0.03 | 0.71±0.48 |
| DKT + Spectral | 0.10±0.01 | 0.03±0.13 | 0.07±0.05 | 0.00±0.09 |
| DKT + NN Linear | 0.04±0.03 | -0.12±0.12 | 0.12±0.05 | 0.30±0.51 |
| NGGP + NN Linear | 0.02±0.02 | -0.47±0.32 | 0.06±0.05 | 0.24±0.91 |
| NGGP + Spectral | **0.03±0.03** | **-0.68±0.23** | **0.03±0.03** | **-0.62±0.24** |

**Object pose prediction** We also study the behavior of NGGP in a pose prediction dataset introduced in [54]. Each task in this dataset consists of 30 gray-scale images with resolution $128 \times 128$, divided evenly into support and query. The tasks are created by selecting an object from the Pascal 3D [51] dataset, rendering it in 100 random orientations, and sampling out of it 30 representations. The goal is to predict the orientation relative to a fixed canonical pose. Note that 50 randomly selected objects are used to create the meta-training dataset, while the remaining 15 are utilized to create a distinct meta-test set. Since the number of objects in meta-training is small, a model could memorize the canonical pose of each object and then use it to predict the target value, completely disregarding the support points during the inference. This would lead to poor performance on the unseen objects in the meta-test tasks. This special case of overfitting is known as the *memorization problem* [54].

We analyze the performance of GP-based models in this setting by evaluating the performance of DKT and NGGP models[3]. We compare them against the methods used in [54], namely MAML [9],

---

[3]Information about architecture and training regime is given in Supplementary Materials A.

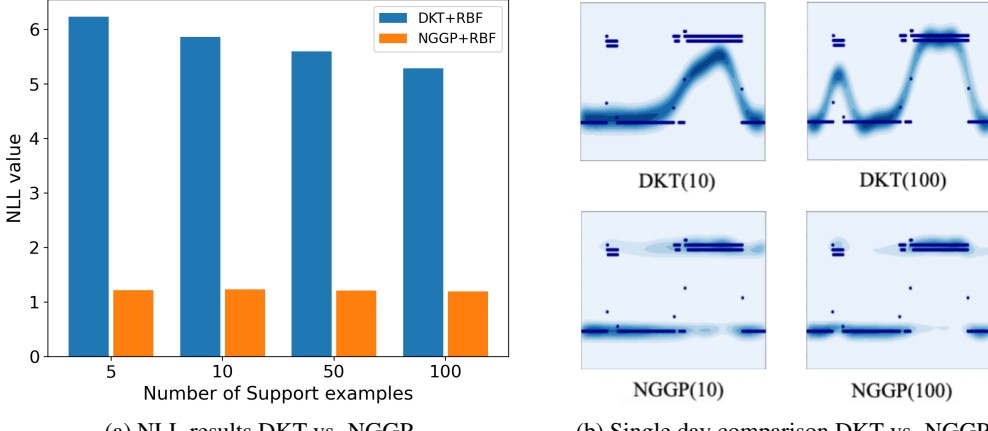

(a) NLL results DKT vs. NGGP.  (b) Single day comparison DKT vs. NGGP.

Figure 5: The results for the Power dataset experiment: **(a)** The quantitative comparison between DKT and NGGP considering different numbers of support examples. **(b)** The power consumption for a single day randomly selected from the test data. We compare DKT vs. NGGP (with RBF kernel) considering 10 and 100 support points. NGGP captures multi-modality and thus better adjusts to the data distribution.

Conditional Neural Processes (CNP) [12] and their meta-regularized versions devised to address the memorization problem — MR-MAML and MR-CNP [54]. In addition, we also include the fine-tuning (FT) baseline and CNP versions with standard regularization techniques such as Bayes-by-Backprop (BbB) [2] and Weight Decay [20]. The results are presented in Table 3.

Both GP-related approaches: NGGP and DKT are similar or usually outperform the standard and meta-regularized methods, which indicates that they are less prone to memorization and therefore benefit from a better generalization. The *NLL* is significantly lower for NGGP than for DKT, confirming that NGGP is better at inferring complex data distributions.

**Power Dataset** In this series of experiments, we use the Power [1] dataset and define an experimental setting for the few-shot setting. We treat each time series composed of 1440 values (60 minutes $\times$ 24 hours) that represents the daily power consumption (*sub_metering_3*) as a single task. We train the model using the tasks from the first 50 days, randomly sampling 10 points per task, while validation tasks are generated by randomly selecting from the following 50 days.

Quantitative and qualitative analysis are provided in Figure 5. We use only *NLL* to assess the results due to the multi-modal nature of the data and analyze the

Table 3: Quantitative results for the object pose prediction task. We report the mean and standard deviation over 5 trials. The lower the better. Asterisks (*) denote values reported in [54].

| Method | MSE | NLL |
|---|---|---|
| MAML* | $5.39 \pm 1.31$ | - |
| MR-MAML* | $2.26 \pm 0.09$ | - |
| CNP* | $8.48 \pm 0.12$ | - |
| MR-CNP* | $2.89 \pm 0.18$ | - |
| FT* | $7.33 \pm 0.35$ | - |
| FT + Weight Decay* | $6.16 \pm 0.12$ | - |
| CNP + Weight Decay* | $6.86 \pm 0.27$ | - |
| CNP + BbB* | $7.73 \pm 0.82$ | - |
| DKT + RBF | $1.82 \pm 0.17$ | $1.35 \pm 0.10$ |
| DKT + Spectral | $\mathbf{1.79 \pm 0.15}$ | $1.30 \pm 0.06$ |
| NGGP + RBF | $1.98 \pm 0.27$ | $\mathbf{0.22 \pm 0.08}$ |
| NGGP + Spectral | $2.34 \pm 0.28$ | $0.86 \pm 0.45$ |

value of the criterion for different numbers of support examples. NGGP better adjusts to the true data distribution, even in the presence of very few support examples during inference. This experiment supports the claim that NGGPs are well-suited for modeling multi-modal distributions and step functions.

**NASDAQ and EEG datasets** In order to test the performance of our methods for real-world time series prediction, we used two datasets - NASDAQ100 [30] and EEG [8]. For an extensive description of the datasets and evaluation regime of this experiment, see Supplementary Materials A. Quantitative results are presented in Table 4. Our experiments show that NGGP outperforms the baseline DKT method across all datasets. The improvement is especially visible for the *out-of-range* NASDAQ100 when both methods use the RBF kernel. The results suggest that NGGPs can be successfully used to model real-world datasets, even when the data does not follow a Gaussian distribution.

Table 4: Quantitative results for NASDAQ and EEG datasets.

(a) NASDAQ100

| in-range | | |
|---|---|---|
| **Method** | *MSE · 100* | *NLL* |
| NGGP + RBF | **0.012 ± 0.014** | **-3.092 ± 0.255** |
| NGGP + NN Linear | 0.023 ± 0.044 | -2.567 ± 1.235 |
| DKT + NN Linear | 0.027 ± 0.032 | -2.429 ± 0.271 |
| DKT + RBF | 0.022 ± 0.042 | -2.878 ± 0.706 |
| out-of-range | | |
| **Method** | *MSE · 100* | *NLL* |
| NGGP + RBF | 0.016 ± 0.034 | -2.978 ± 0.571 |
| NGGP + NN Linear | **0.003 ± 0.004** | **-2.998 ± 0.260** |
| DKT + NN Linear | 0.005 ± 0.006 | -2.612 ± 0.059 |
| DKT + RBF | 0.181 ± 0.089 | 1.049 ± 2.028 |

(b) EEG

| in-range | | |
|---|---|---|
| **Method** | *MSE · 100* | *NLL* |
| NGGP + RBF | **0.222 ± 0.181** | **-1.715 ± 0.282** |
| NGGP + NN Linear | 0.361 ± 0.223 | -1.387 ± 0.273 |
| DKT + NN Linear | 0.288 ± 0.169 | -1.443 ± 0.188 |
| DKT + RBF | 0.258 ± 0.218 | -1.640 ± 0.237 |
| out-of-range | | |
| **Method** | *MSE · 100* | *NLL* |
| NGGP + RBF | 0.463 ± 0.415 | **-1.447 ± 0.221** |
| NGGP + NN Linear | **0.452 ± 0.578** | -1.046 ± 0.624 |
| DKT + NN Linear | 0.528 ± 0.642 | -1.270 ± 0.622 |
| DKT + RBF | 0.941 ± 0.917 | -1.242 ± 0.685 |

## 6 Conclusions

In this work, we introduced NGGP – a generalized probabilistic framework that addresses the main limitations of Gaussian Processes, namely its rigidity in modeling complex distributions. NGGP leverages the flexibility of Normalizing Flows to modulate the posterior predictive distribution of GPs. Our approach offers a robust solution for few-shot regression since it finds a shared set of parameters between consecutive tasks while being adaptable to dissimilarities and domain shifts. We have provided an extensive empirical validation of our method, verifying that it can obtain state-of-the-art performance on a wide range of challenging datasets. In future work, we will focus on applications of few-shot regression problems needing the estimation of exact probability distribution (*e.g.*, continuous object-tracking) and settings where there is a potential discontinuity in similarity for subsequent tasks (*e.g.*, continual learning).

**Limitations** The main limitation of NGGP s is the costs of learning flow-based models, that could be more expensive than using a standard DKT when the data come from a simple distribution. In such a case, other methods like DKT could be more efficient. Moreover, GPs are expensive for tasks with a large number of observations, making NGGP a better fit for few-shot learning rather than bigger settings. Finally, in some cases, it can be more challenging to train and fine-tune NGGP than DKT because the number of parameters and hyper-parameters is overall larger (e.g. the parameters of the flow).

**Broader Impact** Gaussian Processes for regression already have had a huge impact on various real-world applications [7, 53, 21, 25]. NGGPs make it possible to apply *a priori* knowledge and expertise to even more complex real-world systems, providing fair and human-conscious solutions, *i.e.*, in neuroscience or social studies (see experiments on individual power consumption, EEG, and NASDAQ datasets from section 5). The proposed method is efficient and represents a great tool for better uncertainty quantification. Careful consideration of possible applications of our method must be taken into account to minimize any possible societal impact. For instance, the use of NGGP in object-tracking could be harmful if deployed with malevolent and unethical intents in applications involving mass surveillance.

## Acknowledgments

This research was funded by Foundation for Polish Science (grant no POIR.04.04.00-00-14DE/18-00 carried out within the Team-Net program co-financed by the European Union under the European Regional Development Fund) and National Science Centre, Poland (grant no 2020/39/B/ST6/01511). The work of M. Zieba was supported by the National Centre of Science (Poland) Grant No. 2020/37/B/ST6/03463. The work of P. Spurek was supported by the National Centre of Science (Poland) Grant No. 2019/33/B/ST6/00894. This research was funded by the Priority Research Area Digiworld under the program Excellence Initiative – Research University at the Jagiellonian University in Kraków. The authors have applied a CC BY license to any Author Accepted Manuscript (AAM) version arising from this submission, in accordance with the grants' open access conditions.

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
