# A Training Regime

## A.1 Implementation of the GPs

We use the GPyTorch[4] package for the computations of GPs and their kernels. The NN linear kernel is implemented in all experiments as a 1-layer MLP with ReLU activations and hidden dimension 16. For the Spectral Mixture Kernel, we use 4 mixtures.

## A.2 Sines Dataset

For the first experiments on sines functions, we use the dataset from [9]. For each task, the input points $x$ are sampled from the range $[-5, 5]$, and the target values $y$ are obtained by applying $y = A \sin(x - \varphi) + \epsilon$, where the amplitude $A$ and phase $\varphi$ are drawn uniformly at random from ranges $[0.1, 5]$ and $[0, \pi]$, respectively. The noise values $\epsilon$ are modeled by a normal distribution with zero mean and standard deviation equal to $0.1$.

During the training, we use 5 support and 5 query points. The inference is performed over 500 tasks, each consisting of 200 query points and 5 support points. The models are trained for 50000 iterations with batch size 1 (one task per each parameters update) and learning rate 0.001 using the Adam optimizer with $\beta_1 = 0.9$ and $\beta_2 = 0.999$.

The feature extractor for this experiment is implemented by a 2-layer MLP with ReLU activations and hidden dimension 40, which follows the setting of [9]. The last hidden layer is used as the representation for the DKT[5] and NGGP methods in the Gaussian Process framework.

The CNF component for our model was inspired by FFJORD. Our implementation is based on the original code provided by the authors[6]. We use two stacked blocks of CNFs, each composed of two hidden *concatsquash* layers, 64 units each, with *tanh* activation. We adjusted *concatsquash* layers for the conditional variant of CNF by feeding them with an additional conditioning factor - the 40 dim output from the feature extractor.

We use the same settings for the *in-range* heterogeneous noise experiment, but we train the NGGP method for 10000 iterations instead of 50000 since we have noticed that this is enough for the model to converge.

## A.3 Head-pose trajectory

For the head-pose trajectory task, we use the same setting as proposed in [29] with the same feature extractor - convolution neural network with 3 layers, each with 36 output channels, stride 2, and dilation 2. The NN Linear kernel in this experiment is implemented by a 1-layer MLP with ReLU activations and hidden dimension 16.

During the training phase, we use a meta-batch size equal to 5, the learning rate 0.001, and the Adam optimizer with the same configuration as in the sines experiment. Models were trained for 100 iterations. We use 5 support and 5 query points during the train. During the inference, we use 5 points as the support and the remaining samples of the trajectory as the query. We perform the inference over 10 different tasks.

For NGGP, we use the same CNF component architecture as in for the sines dataset. However, we also add Gaussian noise from the Normal distribution $\mathcal{N}(0, 0.1)$ to the head-pose orientations. Adding noise allows for better performance when learning with the CNF component.

## A.4 Object pose prediction

In order to verify the extend of memorization in NGGP, we consider so-called *non-mutually exclusive* tasks. In this setting, the tasks are constructed in such a way that a single model can solve all tasks zero-shot. In particular, we follow the procedure of the pose prediction task introduced in [54]. The few-shot regression dataset is based on the Pascal 3D[7] data [51] and was recreated based on the code

---

[4]https://gpytorch.ai/, available on the MIT Licence

[5]For the DKT implementation we use the code provided at `https://github.com/BayesWatch/deep-kernel-transfer`

[6]`https://github.com/rtqichen/ffjord`

[7]`ftp://cs.stanford.edu/cs/cvgl/PASCAL3D+_release1.1.zip`

from the original research paper [8]. Firstly, the objects were randomly split into the meta-training set (50) and meta-testing (15), then the MuJoCo [44] library was used to render the instances of objects on a table, setting them random orientations. The observation is a tuple consisting of a $128 \times 128$ gray-scale image and its label - orientation relative to a fixed canonical pose. Every task consists of 30 positions sampled from the 100 renderings and divided randomly into *support* and *query*.

During the training, we use a meta-batch of 10 tasks. The NGGP and DKT models were trained over 1000 iterations, with learning rates equal to 0.01 for the kernel parameters, 0.01 for the feature extractor parameters, and 0.001 for the ODE-mapping component. We used the Adam optimizer with the same $\beta$ configuration as in the sines experiment. We also use the same CNF component architecture as in the sines dataset. Similarly, as in the head-pose trajectory experiment, we add Gaussian noise from $\mathcal{N}(0, 0.1)$ to the orientations for better performance . The inference is performed over 100 tasks, which also consist of 15 support and 15 query points. As the feature extractor, we use one of the architectures tested in the original research paper [54] - the convolutional encoder with five layers stacked as follows: 2 convolutional layers with stride 2 and output dimensions 32 and 48; *max pooling layer* with kernel $2 \times 2$; convolutional layer with output dimension 64; *flatten layer* and *linear layer* with output dimension equal to 64.

For this dataset, we tested NGGP and DKT models with RBF and Spectral kernels only. This choice was due to the similarity between head-pose trajectory and object pose prediction settings, and the results show that these two kernels performed the best on such tasks.

## A.5 Power Dataset

The Power Dataset[9] is an UCI benchmark that describes individual household electric power consumption. The original data is composed of 7 time-dependent attributes, but we focus only on the *sub_metering_3* attribute in our experiments. We split the dataset into tasks, where each of the tasks corresponds to daily electricity consumption and is represented by 1440 measurements (in minutes). We train the model using the first 50 days and validate it using the next 50 days. We used the same architecture as for the sines dataset in our experiments, except the feature extractor returns $1D$ embedding.

## A.6 NASDAQ100 and EEG Datasets

The NASDAQ100[10] dataset consists of 81 major stocks under the NASDAQ 100 index. We decided to use the NASDAQ100 dataset with padding that includes 390 points per day over a 105 days interval.

We use 70% of the initial data points of the NDX100 index for the creation of meta-train tasks. The *in-range* meta-tasks were obtained from the last 30% of the data, while the *out-of-range* inference was obtained from the whole time-series of a different stock index. For this purpose, we utilize the time-series given by the YHOO index, which was not used during the training.

The EEG[11] dataset contains raw time series of brainwave signals sampled at 128Hz for 14 electrodes placed at different areas of the patient scalp. Particular patients had been stimulated for various periods, so the time series had different lengths.

The meta-training tasks were obtained form patient $A001SB1\_1$ and electrode $AF4$ from the first 70% of that time-series data points. Same as in NASDAQ100, meta-test tasks were for the *in-range* scenario were obtained from the last 30% of the same data. The *out-of-range* inference tasks were computed on different patient time-series of EEG data points - we used the $A003SB1\_1$ patient.

For both models, we used the same backbone architecture with Adam optimizer parameters set to the same values as in the experiment on the sines dataset with a learning rate set to 0.001. During the training and testing, we used 5 support and 5 query points. The support and query points where

---

[8] https://github.com/google-research/google-research/tree/master/meta_learning_without_memorization, on Apache-2.0 License

[9] https://archive.ics.uci.edu/ml/datasets/individual+household+electric+power+consumption, made available under the "Creative Commons Attribution 4.0 International (CC BY 4.0)" license.

[10] https://cseweb.ucsd.edu/~yaq007/NASDAQ100_stock_data.html

[11] https://archive.ics.uci.edu/ml/datasets/EEG+Steady-State+Visual+Evoked+Potential+Signals, UCI repository dataset

sampled as an random interval of 10 consecutive points. Models were trained with a batch size 1 for 1000 iterations.

## B   Additional Results: Sines Regression

In addition to the GP-based methods reported in the main text, we also summarize the performance of other baseline algorithms on the sines dataset with standard Gaussian noise. The results are presented in Table 5. It may be observed that the DKT and NGGP significantly outperform other approaches. Therefore we only provide a comparison between those two methods in section 5 in the main paper.

Table 5: The *MSE* and *NLL* results for the inference tasks on sines datasets in the *in-range* and *out-range* settings. The lowest results in bold. Asterisks (*) and (**) denote values reported in [45] and [29], respectively. The lower the result, the better.

| Method | in-range | | out-of-range | |
|---|---|---|---|---|
| | *MSE* | *NLL* | *MSE* | *NLL* |
| ADKL* | 0.14 | - | - | - |
| R2-D2* | 0.46 | - | - | - |
| ALPaCA** | 0.14±0.09 | - | 5.92±0.11 | - |
| Feature Transfer/1** | 2.94±0.16 | - | 6.13±0.76 | - |
| Feature Transfer/100** | 2.67±0.15 | - | 6.94±0.97 | - |
| MAML (1 step)** | 2.76±0.06 | - | 8.45±0.25 | - |
| DKT + RBF | 1.36±1.64 | -0.76±0.06 | 2.94±2.70 | -0.69±0.06 |
| DKT + Spectral | **0.02±0.01** | **-0.83±0.03** | 0.04±0.03 | -0.70±0.14 |
| DKT + NN Linear | 0.02±0.02 | -0.73±0.11 | 6.61±31.63 | 38.38±40.16 |
| NGGP + RBF | 1.02±1.40 | -0.74±0.07 | 3.02±2.53 | -0.65±0.08 |
| NGGP + Spectral | **0.02±0.01** | -0.83±0.05 | **0.03±0.02** | **-0.80±0.07** |
| NGGP + NN Linear | 0.04±0.03 | -0.73±0.10 | 7.34±12.85 | 29.86±27.97 |

## C   Additional Results: Classical Regression Tasks

Our main goal was to show improvement of NGGP over standard GPs in the case of a few-shot regression task. Albeit, we test our method also in classical regression task setting. Intuition is that NGGP may be superior to standard GPs in a simple regression setting for datasets with non-gaussian characteristics, but do not expect any improvement otherwise.

### C.1   Classical Regression Tasks

Following the experiments from [23, 40], we decided to run NGGP on regular regression tasks. In this setting, we trained models over 10000 iterations on samples containing 100 points from a given dataset. Averaged results on 500 test samples containing 40 points that were not seen during the training - are presented in 6.

Table 6: Results on classical regression tasks on proposed datasets are inconclusive. One may see that results of methods performance vary between datasets.

| Dataset | *abalone* | | *Ailerons* | | *creeprupt* | |
|---|---|---|---|---|---|---|
| | MSE | NLL | MSE | NLL | MSE | NLL |
| GP + RBF | 1.26 ± 0.68 | **-1.47 ± 0.20** | 1.28 ± 0.66 | -1.47 ± 0.19 | 1.26 ± 0.68 | **-1.47 ± 0.19** |
| DKT + RBF | **1.18 ± 0.28** | -1.41 ± 0.09 | 1.41 ± 0.39 | **-1.49 ± 0.12** | 1.24 ± 0.39 | -1.44 ± 0.12 |
| NGGP + RBF | 1.23 ± 0.29 | -1.44 ± 0.09 | **1.25 ± 0.31** | -1.44 ± 0.10 | **1.10 ± 0.25** | -1.41 ± 0.08 |

### C.2   Sines

Table 7: One may observe that addition of CNF significantly improves results of the classical GP with RBF kernel in such setting.

| | GP + RBF | DKT + RBF | NGGP + RBF |
|---|---|---|---|
| MSE | 1.06 ± 0.24 | 0.72 ± 0.32 | **0.34 ± 0.22** |
| NLL | -0.98 ± 0.10 | -1.20 ± 0.15 | **-1.33 ± 0.13** |

We ran additional experiments on a synthetic dataset of 2d sine waves (as in the setting from Figure 1). The data was generated by randomly sampling either $\sin(x)$ or $-\sin(x)$ for a given point $x$, together with adding uniform noise from $(0.1, 0.5)$. Models were trained for $10000$ iterations over samples from the range $(-5.0, 5.0)$ with $100$ points in one sample. The prediction was done for samples from the interval $(5.0, 10.0)$ - MSE and NLL were averaged on $500$ test samples. We present the quantitative results in Table 7.