# OpenReview forum: "Non-Gaussian Gaussian Processes for Few-Shot Regression"
_NeurIPS.cc/2021/Conference — NeurIPS 2021 Poster_

### Official Review · Reviewer_apVw · 2021-07-12

**Rating:** 7
**Confidence:** 4

**Summary:**

The paper proposes to improve GP for few shot learning by using continuous normalising flows to give non-Gaussian responses from the Gaussian responses given by GP. The approach is illustrated over a range of data sets.

**Limitations And Societal Impact:**

See Q6 above.

**Main Review:**

The paper uses continuous normalising flows (CNF) to to warp the responses of Gaussian processes. This is a significant generalisation of Warped GP [Snelson, 2003], riding on current default use of GP that gives vector responses. The paper is clearly written.

# Detail comments
1) The paper claims that "GP assume a high similarity between subsequent tasks" (lines 30, 136). Some support for this claim, either by way of illustration or references, will be very helpful to motivate this work for few-shot learning.
2) Cite and discuss relation to [Snelson, 2003] and  [Lázaro-Gredilla, 2012].
3) Cite a paper for spectral mixture kernel (line 214).
4) "Continuous normalising flows" should be in the title.
5) In table 3, NGGP+RBF does better than NGGP+Spectral. An analysis of the results will be useful. In particular, for this case, I would like to know if the use of NGGP does away with the need for covariance functions more complicated then RBF/squared-exponential.
6) In limitations, can you also discuss in relation to sparse GP?
7) In addition to lines 300 to 301, it will be useful to give concrete numbers and timing for the experiments that are illustrated in section 5.
8) Lastly, this paper is set within the context of few-shot regression. However, NGGP could be more useful than just for this. Have the authors tried to use NGGP for, say, just normal regression tasks, or quantile regression? How about experiments tried in [Snelson, 2003] and  [Lázaro-Gredilla, 2012]?

Edward Snelson, Zoubin Ghahramani, Carl Rasmussen. Warped Gaussian Processes.  Advances in Neural Information Processing Systems 16 (NIPS 2003)

Miguel Lázaro-Gredilla. Bayesian Warped Gaussian Processes. Advances in Neural Information Processing Systems 25 (NIPS 2012)





**Time Spent Reviewing:**

3

---

> ### Author Response · Authors · 2021-08-09
> **Response**
>
> We thank the Reviewer for the insightful comments about our work. Below we have provided a detailed response to all the comments.
>
> **Comment**: “The paper claims that <<GP assume a high similarity between subsequent tasks>> (lines 30, 136). Some support for this claim, either by way of illustration or references, will be very helpful to motivate this work for few-shot learning.”
>
> Response: In the standard GP setting, one would fit the GP hyper-parameters (i.e., kernel parameters) to each task. Such an approach takes into consideration only local information about the task, disregarding the benefits that might come from analyzing the tasks jointly. Using a shared prior, like in DKT (Patacchiola et al. (NeurIPS 2020)), it is possible to capture the task common information by sharing the deep kernel parameters over all tasks. However, due to the reduced flexibility of the GP model, learning such parameters for tasks with high dissimilarities may be cumbersome. By introducing CNFs, we retain the advantages of a shared prior while being able to overcome the issue of flexibility. As a result, our NGGPs can better adapt to the underlying distributions.
>
> From an empirical perspective, the adaptation to dissimilarities and domain shifts is represented by the out-of-range experiments. In such a setting, at inference time, the model receives tasks for ranges it has never seen during training. This means that the distribution used at inference time can differ from the one used during the training. In such cases, NGGP clearly outperforms other approaches (see especially Table 2 and Table 4 in the paper).
>
> **Comment**: “Cite and discuss relation to [Snelson, 2003] and [Lázaro-Gredilla, 2012].”
>
> Response: We thank the Reviewer for pointing out the papers about Warped Gaussian Processes (Snelson et al. (2003), Lázaro-Gredilla (2012)). We will cite and discuss them in a new paragraph in the Related Work section. Below we shortly discuss these works and their relation to our NGGP method:
>
> Snelson et al. (2003). extends the flexibility of GPs by allowing to process the targets by a learnable monotonic mapping (the warping function). This idea is further extended in Lázaro-Gredilla (2012), who show that it is possible to place another GP prior to the warping function itself. In our approach, the likelihood transformation is obtained by the use of the learnable CNF mapping. However, note that our choice of the mapping and the way it is implemented is specifically tailored for the few-shot regression case, not standard GP regression problems.
>
> **Comment**: “​​Cite a paper for spectral mixture kernel (line 214).”
>
> Response: We thank the Reviewer for pointing this out. We cited the paper for spectral mixture kernel (Wilson & Adams (2013)) at the first appearance (line 99), we will definitely add the citation also in line 214.
>
> **Comment**: “<<Continuous normalising flows>> should be in the title.”
>
> Response: We thank the Reviewer for the suggestion. We agree that the title will be more precise, but we are also afraid it could be too wordy. We will consider changing the title in the final version.
>
> **Comment**: “In table 3, NGGP+RBF does better than NGGP+Spectral. An analysis of the results will be useful. In particular, for this case, I would like to know if the use of NGGP does away with the need for covariance functions more complicated then RBF/squared-exponential.”
>
> Response: In general, the choice of the GP kernel can by itself introduce some bias. In particular, the Spectral kernel tends to work better with periodic functions (see, for instance, Patacchiola et al. (NIPS 2020)), which is also clearly visible in our sines experiments. Note that our NGGP framework allows for the use of any valid kernel. That being said, NGGPs should be less sensitive to misfitted kernels (i.e., kernels that might have been chosen due to some prior believes that do not necessarily hold for a given experiment), since in some cases, the flexibility of the CNF transformation of the targets can still make up for this by suitable transforming the targets. In other words, more complicated covariance functions may be used in NGGPs. In general, we expect the NGGPs to be less sensitive to the choice of these functions than the purely GP-based models.
>
>
> **Comment**: “In limitations, can you also discuss in relation to sparse GP?”
>
> Response: We thank the Reviewer for suggesting adding a discussion about sparse GPs. Sparse GP approaches may significantly reduce the computational cost of GPs with respect to the number of input points, improving the flexibility (for instance, Snelson & Ghahramani (2006)). However,  in this paper, we consider the few-shot learning problem, which means that, in general, we do not expect the number of samples to be high. The computational overhead of NGGPs is not dominated by the GP operations but rather comes from the use of CNFs. In the revised version, we will definitely add the appropriate discussion in the Limitations and related work section regarding the sparse GP approaches.
>
> **Comment**: “In addition to lines 300 to 301, it will be useful to give concrete numbers and timing for the experiments that are illustrated in section 5.”
>
> Response: We will add this in the revised version of the Appendix.
>
> **Comment**: “Lastly, this paper is set within the context of few-shot regression. However, NGGP could be more useful than just for this. Have the authors tried to use NGGP for, say, just normal regression tasks, or quantile regression? How about experiments tried in [Snelson, 2003] and [Lázaro-Gredilla, 2012]?”
>
> Response: Our main aim was to prove the improvement of NGGP over standard GPs in the few-shot regression case. We speculate that NGGPs may be superior to standard GPs in a simple regression setting for datasets with highly non-gaussian characteristics but do not expect any improvement otherwise. Specifically, we observe the better performance of NGGPs compared to GPs in an experiment we have run for the rebuttal. We used the same double sine-wave dataset used to produce Figure 1 in the paper. On this dataset, our NGGP approach is able to achieve an MSE of 0.34+-0.22, while the standard GP is significantly worse with an MSE of 1.06 +- 0.24. GPs are not able to model a multimodal distribution and collapse to predict the mean of those two sines waves. We will add quantitative results for this experiment in the revised version of the paper.
>
>
> **References**:
>
> Lázaro-Gredilla, M. (2012). Bayesian Warped Gaussian Processes. Advances in Neural Information Processing Systems, 25, 1619-1627.
>
> Snelson, E., & Ghahramani, Z. (2006). Sparse Gaussian Processes using Pseudo-inputs. Advances in Neural Information Processing Systems, 18, 1257.
>
> Snelson, E., Rasmussen, C. E., & Ghahramani, Z. (2003). Warped Gaussian Processes. Advances in Neural Information Processing Systems, 16, 337-344.
>
> Wilson, A., & Adams, R. (2013, May). Gaussian Process Kernels for Pattern Discovery and Extrapolation. In International Conference on Machine Learning (pp. 1067-1075). PMLR.

---

> > ### Comment · Reviewer_apVw · 2021-08-30
> > **Few-shot learning (pt 8)**
> >
> > Thanks. I understand the motivation for few-shot learning. However, in the paper itself, sections 4.1 and 4.2 are rather general, and it is section 4.3 that tailors towards few-shot learning. This then translates to Algorithm 1 which was implemented.
> >
> > It will be illustrative nevertheless to have tried NNGP on typical regression problems (i.e., UCI datasets, those examples in the Snelson, 2003 and Lázaro-Gredilla, 2012) to allow the readers to have some idea of the utility of NNGP in typical regression settings. The detail resuls can be in the supplementary material, with a sentence or two as a remark in the main paper.
> >
> > In any case, this is a worthwhile paper, and I maintain my original assessment.

---

> > > ### Author Response · Authors · 2021-09-01
> > > **Response to reviewer's comment**
> > >
> > > Thank you for your response to our rebuttal and further suggestions regarding additional experiments on the standard regression examples.
> > > As you noticed, even if we fit our approach for the few-shot learning setting, sections 4.1 and 4.2 in their generality allow for applying NGGP in the standard regression as well. We are now working on fitting the NGGP in the standard regression tasks - from those already suggested. We are going to include the results in the camera-ready version.
> > > We thank you for acknowledging that our paper is worthwhile, we really appreciate this.

---

### Official Review · Reviewer_3o7L · 2021-07-14

**Rating:** 5
**Confidence:** 4

**Summary:**

Authors propose a method for incorporating continuous normalising flows (CNFs) into the learning mechanism of Gaussian processes (GP). The work is motivated by the scenario of few-shot learning, where they need more flexibility to model subsequent tasks that might differ between them. Experiments show that training such models is possible.

**Limitations And Societal Impact:**

Both limitations and societal impact considerations are well addressed.

**Main Review:**

**Originality and significance:**

- I find that the authors do not cite or refer the work of Maroñas et al. in AISTATS 2021 “Transforming Gaussian Processes with Normalizing Flows”, which is extremely similar to the proposed approach or at least relevant for the analysis or comparison. The AISTATS proceedings were published on the 29th March, so it is not the case of overlapping dates with the submission. Additionally, if one googles “Gaussian Processes + Normalising Flows”, 4/5 results point that work… So I see it very difficult to ignore.
- The proposition of CNFs is interesting to leverage the flexibility of the GP model, I like that. However I do not see clearly the contribution. I understand that it might be to use the ODE-based mechanism of Grathwohl et al. (ICLR 2019) instead of the usual chain of transformations in NFs, but without having cited “Maroñas et al. in AISTATS 2021”, it is difficult to claim.
- Overall, I think that the paper is in the good direction, but I think that its significance is limited as long as some key references are omitted and it is not very well placed in the context of GP models (I see that references to Deep Kernel Learning for few-shot learning are preferred).
- The few-shot learning scenario is not fully motivated, for instance, it is difficult to understand the limitations that it produces in the GP approach; at least to me.

**Quality and clarity:**

- Section 4 introduces a lot of decisions, particularly, on how the proposed model differs from the usual GP learning mechanism. Some of these are not motivated or explained. I consider very important to explain very well each one of them. For instance, why the feature extractor is introduce in the kernel? Is the dimensionality of X super huge? A question arise to me on this point, does this feature extractor preserves the Euclidean distance between data points? If not, what would be the effect on the learning process?
- The notation could be a bit more rigorous, I see that the definition of different variables is equal (for example, the function of the GP and the functions of the NFs). Also in L164 when you refer to Eq. (8) using the transformed variable z, the other equation is written using y. This slightly reduce the clarity and technical quality of the paper..
- The experiments are large enough and I see the effort of authors for demonstrating the performance of the model. However I feel that the comparison with the DKT model is a bit “unfair” or at least reduced. Can be the NLL used to compare different models? My intuition is that it is not allowed.
- The experiment with the sines does not look as a critical situation where a GP could fail, and Figure 5(b) seems a bit overfitted to the data. Would this model generalise once trained? It seems that the DKT preserved better the transition of probability between the examples.

**Questions and other comments:**

- If one looks to ODEs or Grathwohl et al. (ICLR 2019), it is well-known that they are used for modelling dynamical systems. That’s why the time “t” appears and it is integrated in Equation (6). The apparition of the time in the context of the paper is not clear to me, why an ODE is needed? I guess that the “t” variable/dynamics substitute the chain of functional transformations of NFs. Is that right?

**References:**

- J. Maroñas, O. Hamelijnck, J. Knoblauch and T. Damoulas. Transforming Gaussian Processes with Normalizing Flows. AISTATS 2021
- W. Grathwohl, R. T. Q. Chen, J. Bettencourt, I. Sutskever and D. Duvenaud. FFJORD: Free-form Continuous Dynamics for Scalable Reversible Generative Models. ICLR 2019


**Time Spent Reviewing:**

3,5

---

> ### Author Response · Authors · 2021-08-06
> **Response part 1/2**
>
> We thank the Reviewer for pointing out the work of Maroñas et al. (AISTATS 2021); it is a valuable related work that would be deeply discussed in the revised version if accepted. The main idea of our work is similar in spirit (enriching GP distributions with flow-based transformations), but the application domain (our model is focused on few-shot probabilistic regression) and the proposed approach are totally different (due to the different application setting).
>
> Our work follows the family of methods within the few-shot learning setting that utilize a probabilistic framework, e.g., Patacchiola et al. (NIPS 2020). Therefore, our goal was to adapt GPs and powerful Continuous Normalizing Flow models into few-shot problems.
>
> Most importantly, our model (NGGP) and TGP (Maroñas et al. (AISTATS 2021) are not directly comparable. Our model is hard to train on a single large dataset since the flow requires a large number of small datasets (tasks) to train.  On the other hand, the model of Maroñas et al. (AISTATS 2021) cannot be trained on many small tasks since the authors assume that a single large dataset that cannot be changed.  Below we provide a more detailed description of the differences between the two methods.
>
> 1. In few-shot scenarios, we optimize the marginal distribution using only 10-20 examples per task during the inference having only a few support examples (with known inputs and outputs). We aim to estimate the true posterior values for query examples (with known inputs). For our approach, it is crucial to find the true posterior that can be calculated for a varying number of support examples. Practically, it means that we need marginal and posterior in a closed-form. It is possible to achieve this with GPs and NGGPs (our method) but not possible with TGPs (Maroñas et al.  2021). In our method, we obtain a closed-form expression by using the transformation of the components of y, sharing the parameters of the transformation independently among the components. The relation between them is captured by base distribution represented by the GP and the deep kernels.
>
> 2. The architecture used in TGPs and ours differ significantly. Maroñas et al. (2021) use a neural network that takes the input vector and produces the parameters of the flow, but they do not use any deep kernel (see Figure 13 in Maroñas et al. (2021)). We think that using a deep kernel is crucial in many settings, particularly in few-shot learning. In other words, the method of Maroñas et al. (2021) cannot easily account for a deep kernel while it is built-in in our framework. Another architectural difference is that we use the deep kernel embeddings as a conditioner for the flow, while Maroñas et al. (2021) use the inputs. Our approach scales more easily since the dimensionality of the embeddings can be controlled while the dimensionality of the input cannot.
>
> 3. Last but not least, differently from Maroñas et al. (2021), we use an ODE mapping as transformation, which is better at operating on low dimensional data (here, we apply it to 1D vectors coming from the deep kernel) and shares the parameters across all variables, so it is flexible enough to tackle that issue.
>
> Because of the limited space, we provide detailed responses to the rest of the Reviewer’s comments in the following message.

---

> ### Author Response · Authors · 2021-08-06
> **Response part 2/2**
>
> In the next paragraphs we will respond to the rest of the Reviewer’s comments:
>
> **Comment**: “Overall, I think that the paper is in the good direction, but I think that its significance is limited as long as some key references are omitted and it is not very well placed in the context of GP models…”
>
> Response: We will improve the related work section to discuss the difference with other GP methods. However, we would like to point out that the main focus of the paper is on few-shot learning, therefore a comparison with other methods outside of this setting is marginal.
>
>
>
> **Comment**: “The few-shot learning scenario is not fully motivated...”
>
> Response: We will improve the clarity of the paper in the final version, here we provide a short explanation of the limitations of standard GPs. The importance of using our approach VS standard GPs is tightly connected with the way tasks are constructed in few-shot learning. Consider few-shot tasks, as i.i.d. samples from a task distribution. In probabilistic terms, this setting can be represented by defining a set of task-specific parameters (unique to each task) and a set of task-common parameters (shared across all tasks). The way standard GPs have been applied to this setting is to fit the GP hyper-parameters over each task. This approach has several limitations since it only captures the local structure of the task, disregarding any common information shared across all tasks. With our approach instead, we are able to capture the task-common information since we share the GP kernel across tasks, and additionally, we can flexibly model task-specific parameters thanks to the Normalizing Flow. Essentially, standard GPs use a classical Bayesian approach while we are using a hierarchical Bayesian model supplemented with flows.
>
>
> **Comment**: “Section 4 introduces a lot of decisions...”
>
> Response: We will improve Section 4 and clarify some of those points. Regarding the use of deep kernels, we follow a well-established line of work (Hinton and Salakhutdinov (2008); Wilson et al. (2016)). The Reviewer is right regarding the dimensionality of the inputs, deep kernels have been widely used in the GP context for dimensionality reduction. The use of a deep kernel does not affect the GP down the line. We refer the reader to Hinton and Salakhutdinov (2008) and Wilson et al. (2016) for an empirical and theoretical analysis of deep kernels.
>
> **Comment**: “The notation could be a bit more rigorous...”
>
> Response: Yes, we will correct this to be consistent with previous notation.
>
> **Comment**: “However I feel that the comparison with the DKT model is a bit <<unfair>>...”
>
> Response: We did not intend to introduce any unfairness in our evaluation and do not see any, even after a careful read of all reviews of our submission. We have done our best to keep the conditions uniform across experiments (e.g., same deep kernels, same training schedule, etc.), and we think the results are quite robust from this point of view. Regarding the use of the NLL for comparing methods, this is a metric that the community has widely used, especially to compare probabilistic methods in the few-shot learning context (see, for instance, Trippe & Turner (2018), Chai et al. (2019), Grathwohl et al. (ICLR 2019), Tang & Salakhutdinov (2019), Jain et al. (2020, May)). Common choices in statistics for comparing different models are often the Bayesian information criterion (BIC) or Akaike Information Criterion (AIC), which takes under consideration the number of parameters used in the model. However,  in deep learning scenarios, we are dealing with many parameters (e.g., neural net weights), and models can overfit the data completely. In this context, reporting the NLL on the test data is the best choice.
>
> **Comment**: “The experiment with the sines...”
>
> Response: The experiment in Figure 5 (b) refers to the Power dataset. The provided plots are for the test split. The goal of this experiment was to show how NGGP deals with multimodal cases compared to DKT. By comparing the case with 10 support points to the case with 100 support points we aimed at showing that the base distribution of the model (posterior $p(\\mathbf{z}_{*}^{\\mathbf{h}}|\\mathbf{X},\\mathbf{z}^{\\mathbf{h}}, \\mathbf{X}, \\boldsymbol{\\phi}, \\boldsymbol{\\theta})$)
>
> is not collapsing to a single distribution, with the flow component doing most of the job. The experiment shows that by increasing the number of support examples the NGGP becomes more confident in some of the decision regions - therefore the posterior $p(\\mathbf{z}_{*}^{\\mathbf{h}}|\\mathbf{X}, \\mathbf{z}^{\\mathbf{h}}, \\mathbf{X}, \\boldsymbol{\\phi}, \\boldsymbol{\\theta})$ is not collapsing.
>
>
> **Comment**: “If one looks to ODEs or Grathwohl et al. (ICLR 2019)...”
>
> Response: The Reviewer is correct that t is a variable that substitutes the chain of functional transformations of NFs. In sec. 3.2, we describe the idea of CNFs. In general we assume some time-dependent variable $\mathbf{z}(t)$ and function $\mathbf{g}_{\boldsymbol{\beta}}(\mathbf{z}(t), t)$ that models the dynamics of z, $\frac{\partial \mathbf{z}(t)}{\partial t}$. The intuition is that we start from the initial state $\mathbf{z}(t_0)$ that represents the variable described by some well-defined base distribution (in our case, it comes from GP, for Ffjord it is N(0,1)) and finish at final state $\mathbf{z}(t_1)$ that represents the variable from data space with unknown density function (see eq. 6). $t_0$ is usually set to 0. Regarding $t_1$, it is either set to 1 or is trained together with the other parameters that represent the ODE transformation which may speed up the training procedure. From the theoretical perspective, the selection of the values for $t_0$ and $t_1$ is not so important because they are vectorized and processed by a special kind of parametrized layers (concatSquash) in the function of dynamics. The parameters of such layers should be able to scale the time vector properly. However, treating  $t_1$ as an additional parameter adds one more degree of freedom and speeds up the training in practice.
>
> **References:**
>
> Chai, Y., Sapp, B., Bansal, M., & Anguelov, D. (2019). Multipath: Multiple probabilistic anchor trajectory hypotheses for behavior prediction. arXiv preprint arXiv:1910.05449.
>
> Grathwohl, W., Chen, R. T., Bettencourt, J., Sutskever, I., & Duvenaud, D. (2018). Ffjord: Free-form continuous dynamics for scalable reversible generative models. arXiv preprint arXiv:1810.01367.
>
> Hinton, G. and Salakhutdinov, R. (2008). Using deep belief nets to learn covariance kernels for gaussian processes. In Advances in Neural Information Processing Systems.
>
> Jain, A., Casas, S., Liao, R., Xiong, Y., Feng, S., Segal, S., & Urtasun, R. (2020, May). Discrete residual flow for probabilistic pedestrian behavior prediction. In Conference on Robot Learning (pp. 407-419). PMLR.
>
> Maroñas, J., Hamelijnck, O., Knoblauch, J., & Damoulas, T. (2021, March). Transforming Gaussian processes with normalizing flows. In International Conference on Artificial Intelligence and Statistics (pp. 1081-1089). PMLR.
>
> Patacchiola, M., Turner, J., Crowley, E.J., O'Boyle, M.F.P. & Storkey, A.J. (2020), Bayesian Meta-Learning for the Few-Shot Setting via Deep Kernels. In Advances in Neural Information Processing Systems 33. NIPS.
>
> Tang, C., & Salakhutdinov, R. R. (2019). Multiple futures prediction. Advances in Neural Information Processing Systems, 32, 15424-15434.
>
> Trippe, B. L., & Turner, R. E. (2018). Conditional density estimation with Bayesian Normalising Flows. arXiv preprint arXiv:1802.04908.
>
> Wilson, A. G., Hu, Z., Salakhutdinov, R., & Xing, E. P. (2016, May). Deep kernel learning. In Artificial intelligence and statistics (pp. 370-378). PMLR.

---

> > ### Comment · Reviewer_3o7L · 2021-08-17
> > **After rebuttal comments**
> >
> > Thanks to the authors for the detailed comments in their response, I appreciate that. I recognise the effort done to clearly explain the main differences between the missing reference that I pointed out (Maroñas et al. AISTATS 2021) and their work. I now see the main points: i) application to few-shot learning, ii) architecture of the model and iii) parameterisation.
> >
> > With respect to my questions, I think that authors addressed them well. Looking to the one referred to the experiments, I would say that the results are good enough, but it is also true that characterisation metrics could be slightly improved.
> >
> > I think that the paper would deserve to be accepted if authors adequately address the points indicated, also from the other reviewers. However, to me, the lack of reference to Maroñas et al. (AISTATS 2021) is a critical issue that I cannot overlook for proposing acceptance, even if there exist key differences between both methods. For this reason, I will raise my score only to 5/10.

---

> > > ### Author Response · Authors · 2021-08-18
> > > **Response to reviewer's comment**
> > >
> > > Thank you for spending your time and reading our rebuttal, we are glad it clarified your questions and that our explanation regarding the similarity with other work was clear. We will be more than happy to provide more clarifications if needed.

---

### Official Review · Reviewer_AvKx · 2021-07-15

**Rating:** 6
**Confidence:** 3

**Summary:**

The authors propose to make the output distributions of Gaussian processes more flexible by combining them with continuous normalizing flows. They then show that in the application of few-shot learning, this can have benefits over standard GPs without the flows.

UPDATE: Score raised due to clarifications in the author response.

**Limitations And Societal Impact:**

The limitations of the method are hard to assess, mostly because the choice of CNFs over NFs or any other flexible distribution family is not well motivated and because (theoretical and empirical) comparisons to many relevant related methods are missing. This should be addressed.

**Main Review:**

Strengths:
- The idea seems to be novel and it is intuitive that NFs could help improve the flexibility of the distribution.
- The empirical results are promising.

Weaknesses:
- The description of the method is somewhat unclear and it is hard to understand all the design choices.
- Some natural baselines and important related work seem to be missing.

Major comments:
- The lack of flexibility of standard GPs is not a new observation as has been approached in the past, possibly most famously by the deep GP [1]. These models have recently become a mainstream tool with easily usable frameworks [e.g., 2], so that it would seems like a natural baseline to compare against.
- Generally, a lot of related work seems to be missed by this paper. For instance, meta-learning kernels for GPs for few-shot tasks has already been done by [3] and then later also by [4,5,6]. These should probably be mentioned and it should be discussed how the proposed method compares against them.
- The paper proposes to use CNFs, but these require solving a complex-looking integral (e.g., Eq. 9). It should be discussed how tractable this integral is or how it is approximated in practice. Moreover, it seems like an easier choice would be standard NFs, so it should be discussed why CNFs are assumed to be better here. Possibly, one should also directly compare against a model with a standard NF as an ablation study.
- In l. 257ff it is claimed that the proposed GP methods are less prone to memorization. How does this compare to the results in [4], where DKT seems to memorize as well? Could the regularization proposed in [4] be combined with the proposed model?

Minor comments:
- In l. 104 it is said that every kernel can be described by a feature space parameterized by a neural network, but this is trivially not true. For instance, for RBF kernels, the RKHS is famously infinite-dimensional, such that one would need an NN with infinite width to represent it. So at most, NNs can represent finite-dimensional RKHSs in practice. This limitation should be made more clear.
- l. 151 with GP -> with a GP
- l. 152 use invertible mapping -> use an invertible mapping
- l. 161 the "marginal log-probability" is more commonly called "log marginal likelihood" or "log evidence"
- Eq. (8): should it be $\mathbf{z}$ instead of $\mathbf{y}$?
- In the tables, it would be more helpful to also bolden the fonts of the entries where the error bars overlap with the best entry.

[1] Damianou & Lawrence 2012, https://arxiv.org/abs/1211.0358

[2] Dutordoir et al. 2021, https://arxiv.org/abs/2104.05674

[3] Fortuin et al. 2019, https://arxiv.org/abs/1901.08098

[4] Rothfuss et al. 2020, https://arxiv.org/abs/2002.05551

[5] Venkitaraman et al. 2020, https://arxiv.org/abs/2006.07212

[6] Titsias et al. 2020, https://arxiv.org/abs/2009.03228

**Time Spent Reviewing:**

3

---

> ### Author Response · Authors · 2021-08-09
> **Response part 1/2**
>
> We thank the Reviewer for the useful comments about our work, we agree with many of them. Below we provide our answer.
>
> **Major comments:**
>
> **Comment**: “The lack of flexibility of standard GPs is not a new observation as has been approached in the past, possibly most famously by the deep GP [1]. These models have recently become a mainstream tool with easily usable frameworks [e.g., 2], so that it would seems like a natural baseline to compare against.”
>
> Response: The lack of flexibility of standard GPs is indeed not a new observation. However, more flexible methods like deep GPs (Damianou & Lawrence (2013)) cannot be easily adapted to the few-shot setting. This is mainly due to the significant differences in the tasks structure and is one of the reasons why deep GPs (Damianou & Lawrence (2013)) are not our natural baseline. Since we are working on GP flexibility in the few-shot regression setting, we have used a state-of-the-art method specifically tailored for this case as our baseline (DKT; Patacchiola et al., (NeurIPS 2020)).
>
> **Comment**: “Generally, a lot of related work seems to be missed by this paper. For instance, meta-learning kernels for GPs for few-shot tasks has already been done by [3] and then later also by [4,5,6]. These should probably be mentioned and it should be discussed how the proposed method compares against them.”
>
> Response: We thank the Reviewer for pointing out those important papers. We will add a separate paragraph in the Related Work section and discuss each one of these papers, showing the differences between these models and our method. Below we shortly summarize the differences:
>
> The work of Fortuin et al. (2019) proposes to learn the mean function of the GP by using the knowledge from multiple tasks in a meta-learning setting. In our approach, the mean function is typically assumed to be zero (a common assumption in the absence of any prior knowledge), and it is the deep kernel to be learned during the meta procedure. In addition, we use a CNF to increase the flexibility of the model. Learning the mean function as in Fortuin et al. (2019) could be easily introduced in our model, this would probably increase the performance even further - we might consider it for future work. We thank the Reviewer for pointing out this paper.
>
> In Rothfuss et al. (2021), the authors presented a theoretically principled PAC-Bayesian framework for meta-learning. It can be used with different base learners (e.g., GPs or BNNs). We believe that our NGGP method could be used within that framework as another base learner type, which produces more flexible target posteriors.
>
> The work of Venkitaraman et al. (2020) also explores topics related to kernel tricks and meta-learning. They propose to use nonparametric kernel regression for the inner loop update. Note that their work is crucially different from ours, as they do not use Gaussian Processes. Moreover, in our case, the kernels are clearly parametric (modeled by deep kernels).
>
> In Titsias et al. (2020),  the authors introduce an information theoretic framework for meta-learning by using a variational approximation to the information bottleneck. In their GP-based approach, to account for likelihoods other than Gaussians, they propose to approximate the non-Gaussian terms in the posterior with Gaussian distributions (by using amortized functions), while we use CNFs to increase the flexibility of the GPs. Similar to us, they heavily rely on the comparison with the DKT method  (Patacchiola et al. (NeurIPS 2020)), but (except for a simple sine dataset), they consider only few-shot classification. This is in contrast with our work since our model is designed specifically for few-shot regression and is not directly applicable to classification.
>
> **Comment**: “The paper proposes to use CNFs, but these require solving a complex-looking integral (e.g., Eq. 9). It should be discussed how tractable this integral is or how it is approximated in practice. Moreover, it seems like an easier choice would be standard NFs, so it should be discussed why CNFs are assumed to be better here. Possibly, one should also directly compare against a model with a standard NF as an ablation study.”
>
> Response: The motivation for selecting CNFs instead of standard NFs is because they allow constructing the model for which marginal and posterior are given in a closed-form. We use an invertible transformation that operates on individual components of the output $\mathbf{y}$ and shares the parameters across them. In practice, the transformation is applied to 1D data. Considering the empirical results provided by the authors of Ffjord (Grathwohl et al. (2018); Table 2) that uses the same transformation, the proposed approach performs better than discrete flows like RealNVP (Dinh et al. (2016)) or Glow (Kingma & Dhariwal (2018)) for low-dimensional data in terms of NLL (we use even one channel from the Power dataset, for which Ffjord behaves significantly better). Note that flows that use coupling layers (RealNVP (Dinh et al. (2016)), Glow (Kingma & Dhariwal (2018))) and autoregressive flows (MAF (Papamakarios et al. (2017))) do not make sense for 1D data.  Since we operate on 1D data, we do not care about simplifying the estimation of the Jacobian, and any invertible differentiable transformation can be applied. At the same time, we need a complex, well-parameterized transformation because we share the parameters across all GP outputs.
>
> To solve the complex-looking integral from eq. (9) we follow the previous methods (Grathwohl et al. (2018)) and use the Dormand–Prince solver (Dormand & Prince (1980))  computing the gradients using the adjoint sensitivity method. We will include this detail in the revised version.
>
> **Comment**: “In l. 257ff it is claimed that the proposed GP methods are less prone to memorization. How does this compare to the results in [4], where DKT seems to memorize as well? Could the regularization proposed in [4] be combined with the proposed model?”
>
> Response: We carefully checked the paper of Rothfuss et al. (2021) pointed out by the Reviewer, but we could not find anything about the DKT method. We notice only that the GPs (not DKT) were indicated as prone to memorization problems. In principle, it should be possible to combine the regularization proposed by the paper of Rothfuss et al. (2021) with our NGGP method - in other words, we think that NGGP could be a base learner for PACOH regularization.
>
>
> Because of the limited space, we provide detailed responses to the rest of the Reviewer's comments in the following message.

---

> ### Author Response · Authors · 2021-08-09
> **Response part 2/2**
>
> In the next paragraphs, we will respond to the rest of the Reviewer's comments.
>
> **Minor comments**:
>
> **Comment**: “In l. 104 it is said that every kernel can be described by a feature space parameterized by a neural network, but this is trivially not true. For instance, for RBF kernels, the RKHS is famously infinite-dimensional, such that one would need an NN with infinite width to represent it. So at most, NNs can represent finite-dimensional RKHSs in practice. This limitation should be made more clear.”
>
> Response: In l. 104, we say that every kernel can be described as an inner product in a feature space - imposed by some feature mapping $\phi$. In some of our experiments, we use a neural network to approximate this mapping. We refer to this implementation as the “NN linear” kernel. Approximating  $\phi$ with a neural network comes with all the limitations (and design choices) of this model. We do not claim that every kernel can be described by a feature space parametrized by a neural network. We will rephrase this sentence to be sure that it is not ambiguous.
>
> **Comment**: “l. 151 with GP -> with a GP; 152 use invertible mapping -> use an invertible mapping”
>
> Response: We will add the missing article in the revised version.
>
> **Comment**: “l. 161 the <<marginal log-probability>> is more commonly called <<log marginal likelihood>> or <<log evidence>>”
>
> Response: We thank the Reviewer for the suggestion. We have seen papers using the first nomenclature and other papers using the second. We agree that log marginal likelihood is more appropriate, and we will use this in the final version.
>
> **Comment**: “Eq. (8): should it be z instead of y?”
>
> Response: Yes, we will correct this to be consistent with the previous notation.
>
> **Comment**: “In the tables, it would be more helpful to also bolden the fonts of the entries where the error bars overlap with the best entry.”
>
> Response: We decided to highlight only the ones with the lowest mean, as this convention was also used in other papers (e.g., Patacchiola et al. (2020)) (see table 3 - results for DKT + RBF/Spectral are in the error interval but only one is bolded). We will consider highlighting the rest in the final version.
>
>
> **Limitations And Societal Impact:**
>
> **Comment**: “The limitations of the method are hard to assess, mostly because the choice of CNFs over NFs or any other flexible distribution family is not well motivated and because (theoretical and empirical) comparisons to many relevant related methods are missing. This should be addressed.”
>
> Response:
> The choice of CNFs over NFs was motivated by the need of obtaining the marginal and the posterior in a closed form, and by the ability of CNFs to work on 1D data (as we describe in detail in response to the comment “The paper proposes to use CNFs, but these require solving…”).
> We thank the Reviewer for pointing out relevant work. We will extend section 2 with another subsection dedicated to those papers (for a more detailed commentary on how these works differ from our model, please refer to the response for  “Generally, a lot of related work seems to be missed…”).
> We addressed other limitations of our method in comparison to other approaches (such as DKT) in “Limitations” (Section 6). We also commented on the broader impact of the work. We will improve those sections if required.
>
>
> **References**:
>
> Damianou, A., & Lawrence, N. D. (2013, April). Deep Gaussian Processes. In Artificial intelligence and statistics (pp. 207-215). PMLR.
>
> Dinh, L., Sohl-Dickstein, J., & Bengio, S. (2016). Density estimation using Real NVP. arXiv preprint arXiv:1605.08803.
>
> Dormand, J. R., & Prince, P. J. (1980). A family of embedded Runge-Kutta formulae. Journal of computational and applied mathematics, 6(1), 19-26.
>
> Fortuin, V., Strathmann, H., & Rätsch, G. (2019). Meta-learning Mean Functions for Gaussian processes. arXiv preprint arXiv:1901.08098.
>
> Grathwohl, W., Chen, R. T., Bettencourt, J., Sutskever, I., & Duvenaud, D. (2018). Ffjord: Free-form Continuous Dynamics for Scalable Reversible Generative Models. arXiv preprint arXiv:1810.01367.
>
> Kingma, D. P., & Dhariwal, P. (2018). Glow: Generative Flow with Invertible 1x1 Convolutions. arXiv preprint arXiv:1807.03039.
>
> Papamakarios, G., Pavlakou, T., & Murray, I. (2017). Masked Autoregressive Flow for Density Estimation. arXiv preprint arXiv:1705.07057.
>
> Patacchiola, M., Turner, J., Crowley, E.J., O'Boyle, M.F.P. & Storkey, A.J. (2020), Bayesian Meta-Learning for the Few-Shot Setting via Deep Kernels. In Advances in Neural Information Processing Systems 33.

---

> > ### Comment · Reviewer_AvKx · 2021-08-16
> > **Thanks**
> >
> > Thanks for the detailed response and clarifications. It is now clearer to me why the CNF was chosen. Regarding the baselines, I agree with your hunch that the different baselines I suggested might not work as well as your method. However, I would still find it more convincing to actually see that confirmed empirically. I will raise my score.

---

> > > ### Author Response · Authors · 2021-08-18
> > > **Response to reviewer's comment**
> > >
> > > Thank you for your response to our rebuttal, we appreciate that it provides clarifications to your questions.

---

### Official Review · Reviewer_SDgP · 2021-07-31

**Rating:** 6
**Confidence:** 3

**Summary:**

The authors propose a novel GP-based model for few-shot regression. The model makes GPs more flexible by adding Continuous Normalizing Flow to the output, specifically an ODE-based mapping. Moreover, this mapping is contextualized by making it input-dependent, which adds more expressivity to the predictive posterior. This flexibility makes the model better suited for modelling complex distributions (e.g. multimodality cases) and, as shown experimentally, is beneficial for few-shot regression, where tasks might differ (domain shift).

The proposed model is experimentally shown to perform better than the main competitor, Deep Kernel Transfer (DKT), on a variety of few-shot regression datasets.

Overall, this is a nicely written paper with an interesting novel method and good experimental results. Some parts of the model, however, are not given enough explanation and testing.

**Limitations And Societal Impact:**

Limitations of the proposed method are described in the paper. Negative societal impact is discussed appropriately given the general theoretical nature of the work.

**Main Review:**

### Strengths:

The prosed methodology is solid. Bayesian non-parametrics, and GP specifically, is very suitable for tackling the task of few-shot regression. Its extension with CNF for more flexibility in the predictive posterior is a very sensible thing to do. To the best of my knowledge, the math is correct.

The method is well-tested for the given task of few-shot regression on 5 datasets.

The paper is well written and clear.

### Weaknesses:

The paper attempts to do two things at once: propose a new model and apply it for a specific task of few-shot regression. Because of this, the model is not described in enough detail. Invertible ODE-based mapping is not described at all. I would like to see some synthetic experiments comparing the proposed model to a GP on a standard regression task, before moving on to few-shot regression experiments.

The two main problems of standard GPs wrt the few-shot learning task, that this work addresses, are limited flexibility of the posterior distribution and the assumption of high similarity between tasks. While the benefit of NGGP in terms of modelling more complex distributions is clearly explained and shown (e.g. the multimodal Power Dataset experiment), the other main claim is not well supported. The authors point that "GPs assume similarity between subsequent tasks" and claim that the proposed method (NGGP) is "adaptive to dissimilarities and domain shifts". However, they do not explain how exactly the proposed method is supposed to adapt to domain shifts. Explaining this as "contextualization provided by the ODE-based mapping" is way too vague, especially given that ODE based mapping is not discussed much at all. Even though the experiments show good performance, it could be attributed to the more flexible posterior. This adaptation claim definitely should be supported by a detailed explanation.



### Other comments and typos:

A flow transformation in the model is applied to the noisy GP outputs. I would like to see some more discussion about this design choice and its potential limitations.

66: "overweights" -> "weights" ?

120: in eq 5 sum is over n, but in the det you put k.

144: ODE-based mapping should be described.

Fig 4: why do you plot different kernels? is it the best-performing ones?

Fig 4 (a) the due to the vertical scaling, we pretty much can not see the predictive mean (red) in the first two plots on the bottom.

**Time Spent Reviewing:**

7

---

> ### Author Response · Authors · 2021-08-10
> **Response**
>
> We thank the Reviewer for the insightful comments about our work, we will respond to them below.
>
> **Weaknesses**
>
> **Comment**: “Because of this, the model is not described in enough detail. Invertible ODE-based mapping is not described at all.”
>
> Response: We will improve the final draft and add more details about the ODE mapping. We are using transformations that are characteristic for continuous normalising flows given by eq. (6). The inverted form of the transformation is as follows:
> $\\mathbf{f}\_{\\boldsymbol{\\beta}}^{-1}( \\mathbf{y})  = \\mathbf{y} - \\int^{t_1}_{t_0} \\mathbf{g}\_{\\boldsymbol{\\beta}}(\\mathbf{z}(t), t) dt$
>
> Because we use a transformation that operates on elements of $\mathbf{y}$ independently and shares the parameters across them, we represent it as:
> $\\mathbf{f}\_{\\boldsymbol{\\beta}}^{-1}(\\mathbf{y})=[f_{\\boldsymbol{\\beta}}^{-1}(y_1),\\dots,f_{\\boldsymbol{\\beta}}^{-1}(y_D)]^{\\mathrm{T}}$
> where:
> $f_{\\boldsymbol{\\beta}}^{-1}(y_d) = y_d - \\int^{t_1}\_{t_0} g_{\\boldsymbol{\\beta}}(z_d(t), t) dt$
>
> In the final model, we use additional information about the input embedding: $\\mathbf{h}\_{\\boldsymbol{\\phi}}(\\mathbf{x}\_d)$. Therefore, for single $y_d$, we use the transformation given by eq. (10) for which the inverse form is:
> $f_{\\boldsymbol{\\beta}}^{-1}(y_d) = y_d -  \\int^{t_1}\_{t_0} g_{\\boldsymbol{\\beta}}(z_d(t), t, \\mathbf{h}_{\\boldsymbol{\\phi}}(\\mathbf{x}_d)) dt$
>
> We are going to clarify all these points in the revised version.
>
>
> **Comment**: “I would like to see some synthetic experiments comparing the proposed model to a GP on a standard regression task, before moving on to few-shot regression experiments.”
>
> Response:  Since we were mainly concerned with the few-shot setting, the proposed structure of our model, the design choices, and the implementation are specially tailored to address this problem. We will include some additional experiments in the standard regression case in the final version. Our intuition is that NGGP may be superior to standard GPs in simple regression settings for datasets with multimodal characteristics that cannot be modeled by using standard Gaussian assumptions. For the rebuttal, we have tried a simple regression dataset with a double sine wave (similar to the one used in Figure 1 in the paper). In this experiment, our approach is able to achieve an MSE of 0.34+-0.22, while the standard GP is significantly worse with a result of 1.06 +- 0.24. The GP is not able to model a multimodal distribution and collapses to predicting the mean of those two sines waves. We will add quantitative results for this experiment in the revised version of the paper.
>
> **Comment**: “The two main problems of standard GPs wrt the few-shot learning task, that this work addresses, are limited flexibility of the posterior distribution and the assumption of high similarity between tasks. While the benefit of NGGP in terms of modelling more complex distributions is clearly explained and shown (e.g. the multimodal Power Dataset experiment), the other main claim is not well supported. The authors point that <<GPs assume similarity between subsequent tasks>> and claim that the proposed method (NGGP) is <<adaptive to dissimilarities and domain shifts>>. However, they do not explain how exactly the proposed method is supposed to adapt to domain shifts. Explaining this as <<contextualization provided by the ODE-based mapping>> is way too vague, especially given that ODE based mapping is not discussed much at all. Even though the experiments show good performance, it could be attributed to the more flexible posterior. This adaptation claim definitely should be supported by a detailed explanation.”
>
> Response: Note that in the standard GP setting, one would fit the GP hyper-parameters (i.e., kernel parameters) to each task. Such an approach takes into consideration only local information about the task, disregarding the benefits that might come from analyzing the tasks jointly. Using a shared prior, like in DKT (Patacchiola et al. (NeurIPS 2020)), it is possible to capture the task common information by sharing the deep kernel parameters over all tasks. However, due to the reduced flexibility of the GP model, learning such parameters for tasks with high dissimilarities may be cumbersome. By introducing CNFs, we retain the advantages of a shared prior while being able to overcome the issue of flexibility. As a result, our NGGPs can better adapt to the underlying distributions.
>
> From an empirical perspective, the adaptation to dissimilarities and domain shifts is represented by the out-of-range experiments. In such a setting, at inference time, the model is given tasks for ranges it has never seen during training. This means that the distribution used at inference time can differ from the one used during the training. In such cases, NGGP clearly outperforms other approaches (see Table 2 and Table 4 in the paper).
>
>
> **Other comments and typos:**
>
> **Comment**: “A flow transformation in the model is applied to the noisy GP outputs. I would like to see some more discussion about this design choice and its potential limitations.”
>
> Response: We apply the flow transformation to noisy outputs because, in real-world scenarios, we have access to a finite dataset. Therefore, training the flow using a limited number of examples without noise injection may lead to collapsing to the expected value of the base distribution. The main limitation of such an approach is the need to select proper variance for the GP noise. However, we follow the approach from DKT, where the variance of GP noise is parameterized and estimated during training together with other parameters.
>
>
> We thank the Reviewer for the minor comments, we will respond to them below.
>
> **Comment**: “66: <<overweights>> -> <<weights>> ?”
>
> Response: There is a typo, it should be “over weights”. We will change it in the manuscript.
>
> **Comment**: “120: in eq 5 sum is over n, but in the det you put k.”
>
> Response: Definitely, it should be a sum over n. We will change it in the revised version.
>
> **Comment**: “144: ODE-based mapping should be described.”
>
> Response: We are going to do this in the revised version.
>
> **Comment**: “Fig 4: why do you plot different kernels? is it the best-performing ones?”
>
> Response: The plotted kernels are the best-performing ones in this experiment, as the Reviewer correctly suggested. We will explicitly mention it in the final version to make it clear.
>
> **Comment**: “Fig 4 (a) the due to the vertical scaling, we pretty much can not see the predictive mean (red) in the first two plots on the bottom.”
>
> Response: We thank the Reviewer for pointing out this graphical issue in the figure, we will change the scaling of the plots in the revised version.
>
>
> **References:**
>
> Patacchiola, M., Turner, J., Crowley, E.J., O'Boyle, M.F.P. & Storkey, A.J. (2020), Bayesian Meta-Learning for the Few-Shot Setting via Deep Kernels. In Advances in Neural Information Processing Systems 33. NIPS.

---

> > ### Comment · Reviewer_SDgP · 2021-09-01
> > **Acknowledging authors' response**
> >
> > Thank you for your response.  You've clarified most of my and other reviewers' questions.
> >
> > Regarding the transformation of the noisy signal, one clear downside is that this design choice limits your model to continuous output likelihoods, meaning your model can not be easily extended to classification tasks.
> >
> > You should definitely discuss the connection to [Maroñas et al. 2021],[Snelson, 2003], and [Lázaro-Gredilla, 2012] in the manuscript, as pointed out by other reviewers.
> >
> > I will keep my rating.

---

> > > ### Author Response · Authors · 2021-09-01
> > > **Response to reviewer’s comment**
> > >
> > > Thank you for the response. We will add a deep discussion with the mentioned papers in the related work section in the revised version of the manuscript.

---

### Author Response · Authors · 2021-08-10
**[Official Comment - To All Reviewers and Area Chairs]**

We thank the Reviewers for their helpful comments. Overall all the Reviewers agreed that our work is novel (e.g., Reviewers: AvKx, 3o7L, apVw, SDgP) and solid (e.g., Reviewer SDgP), with the paper being well written (e.g., Reviewers: SDgP, apVw) and the experiments being thorough (Reviewers SDgP, AvKx). Additionally, the Reviewer apVw mentioned the potential of our NGGP method outside the few-shot learning setting (e.g., in standard regression), pointing out how NGGPs can be considered as a generalization of the well-established Warped GPs (Snelson et al. (2003)). Other reviewers (e.g., SDgP and AvKx) also agree that our idea of exploiting NFs/CNFs can increase the flexibility of the GP predictive posterior, confirming the importance of our main contribution.

Some of the Reviewers suggested additional work to be referenced in our paper. In particular, Reviewer 3o7L pointed out the relevant work of Maroñas et al. (AISTATS 2021), and after a careful read, we identified several differences between this approach and others, presented in detail in our rebuttal response to the Reviewers. Other Reviewers suggested including: Warped GPs, Deep GP, and Sparse GP. Although we agree that these papers should be briefly discussed, we want to emphasize that our work focuses on the particular case of few-shot regression and that all the above-mentioned methods are not designed for this application, and their adaptation to our case requires significant alterations of the baseline algorithms. Therefore, we are not able to include them in the empirical comparison in the current version of the submission, but we will discuss the relation to those works in the camera-ready if accepted and extend the evaluation for the paper extensions. To reiterate, in our paper, we focus mostly on comparing our approach against the recent state-of-the-art GP-based method (DKT), which bears the main similarities to our approach (Patacchiola et al. (NeurIPS 2020)).

Other than the improvements discussed above, we have run additional experiments in the standard regression case to show the effectiveness of our method. Those experiments are briefly described in the comment to Reviewer SDgP. Finally, we will commit to improving the Related Work section and fixing some minor inconsistencies in the notation for the camera-ready.

---

### Decision · Program_Chairs · 2021-09-27

**Decision:**

Accept (Poster)

**Comment:**

The reviewers ultimately agree that the key idea proposed in this paper, namely the use of continuous normalizing flows (CNFs) to achieve highly flexible non-Gaussian predictive posteriors, applicable to few-shot regression and achieving strong results in that domain. This core idea is both natural and novel, and the empirical validation on the particular task of few-shot regression is strong.

While I am ultimately recommending acceptance, I want to echo the comments of a few of the reviewers that the somewhat narrow focus on few-shot learning seems to arbitrarily limit the potential breadth of impact of this paper. This is particularly true since the core methodological contribution -- namely the use of a CNF to achieve a non-Gaussian predictive posterior -- is largely separate from the adaptation to few-shot regression, which involves utilizing simple weight sharing in a deep kernel, alternating optimization over tasks.

Finally, I'll note that because of the relative decoupling of the authors' core contribution and the mechanism they use to approach few shot learning, the argument made in the author feedback that they don't compare to deep GPs etc because these methods are not intended for few-shot regression is somewhat unconvincing. A number of papers have explored the combination of deep kernels with deep GP models, so the adaptation to DKT is quite straightforward. Because deep GPs result in (continuous) mixture predictive distributions, a number of deep GP papers presenting toy results very similar to your own Figure 1 as motivation. I would urge the authors to consider the above points, and how their papers impact may be broadened by including these comparisons.